# Comparative Biomechanical Stability of the Fixation of Different Miniplates in Restorative Laminoplasty after Laminectomy: A Finite Element Study

**DOI:** 10.3390/bioengineering11050519

**Published:** 2024-05-20

**Authors:** Guoyin Liu, Weiqian Huang, Nannan Leng, Peng He, Xin Li, Muliang Lin, Zhonghua Lian, Yong Wang, Jianmin Chen, Weihua Cai

**Affiliations:** 1Department of Orthopedics, The Affiliated Jinling Hospital of Nanjing Medical University, Nanjing 211166, China; liuguoyin0425@163.com (G.L.); huangweiqian2024@163.com (W.H.); 13770794062@163.com (N.L.); yifeiyi_1988@126.com (P.H.); linmuliang@hotmail.com (M.L.); 2Department of Orthopedics, The First Affiliated Hospital of Nanjing Medical University, Nanjing 210029, China; 3Department of Orthopedics, Central Military Commission Joint Logistics Support Force 904th Hospital, Wuxi 214044, China; lixin198611@126.com; 4Xiamen Medical Device Research and Testing Center, Xiamen 361022, China; lianzhonghua@double-medical.com; 5Outpatient Department of The Affiliated Jinling Hospital of Nanjing Medical University, Nanjing 211166, China; wangyong_20232023@163.com

**Keywords:** biomechanics, finite element analysis, laminectomy, restorative laminoplasty, stability

## Abstract

A novel H-shaped miniplate (HSM) was specifically designed for restorative laminoplasties to restore patients’ posterior elements after laminectomies. A validated finite element (FE) model of L2/4 was utilized to create a laminectomy model, as well as three restorative laminoplasty models based on the fixation of different miniplates after a laminectomy (the RL-HSM model, the RL-LSM model, and the RL-THM model). The biomechanical effects of motion and displacement on a laminectomy and restorative laminoplasty with three different shapes for the fixation of miniplates were compared under the same mechanical conditions. This study aimed to validate the biomechanical stability, efficacy, and feasibility of a restorative laminoplasty with the fixation of miniplates post laminectomy. The laminectomy model demonstrated the greatest increase in motion and displacement, especially in axial rotation, followed by extension, flexion, and lateral bending. The restorative laminoplasty was exceptional in preserving the motion and displacement of surgical segments when compared to the intact state. This preservation was particularly evident in lateral bending and flexion/extension, with a slight maintenance efficacy observed in axial rotation. Compared to the laminectomy model, the restorative laminoplasties with the investigated miniplates demonstrated a motion-limiting effect for all directions and resulted in excellent stability levels under axial rotation and flexion/extension. The greatest reduction in motion and displacement was observed in the RL-HSM model, followed by the RL-LSM model and then the RL-THM model. When comparing the fixation of different miniplates in restorative laminoplasties, the HSMs were found to be superior to the LSMs and THMs in maintaining postoperative stability, particularly in axial rotation. The evidence suggests that a restorative laminoplasty with the fixation of miniplates is more effective than a conventional laminectomy due to the biomechanical effects of restoring posterior elements, which helps patients regain motion and limit load displacement responses in the spine after surgery, especially in axial rotation and flexion/extension. Additionally, our evaluation in this research study could benefit from further research and provide a methodological and modeling basis for the design and optimization of restorative laminoplasties.

## 1. Introduction

In neurosurgical and orthopedic practice, patients frequently exhibit symptomatic intraspinal lesions, including intraspinal tumors, hematomas, vascular malformations, and spinal stenosis resulting from hypertrophic facet joints or bulging discs [1,2,3]. Regarding surgical management, a gross total resection to reduce or relieve the compression of the nerve root or spinal cord is considered to be the only effective and timely treatment and has been widely accepted as the standard procedure in clinical practice [4]. From a surgical perspective, a laminectomy is typically performed to provide adequate exposure and the complete removal of lesions within the spinal canal [5]. The surgical treatment of symptomatic intraspinal lesions has gradually evolved from a traditional laminectomy, which entailed the removal of posterior elements, such as the lamina (vertebral plate), spinous processes, and the posterior ligament complex (PLC) [6]. The initial laminectomy was first carried out by Gowers et al. [7] in 1888, typically without the restoration of the excised posterior column. As per the three-column theory of spines proposed by Denis et al. [8], the anterior and middle columns provide support for the spine, while the posterior column safeguards the spinal cord and nerve roots. From a mechanical point of view, the posterior column of the spine bears approximately 24% to 30% of the pressure and 21% to 54% of the rotation pressure of the spine, contributing significantly to its overall stability [9]. When performing a laminectomy, damage to the posterior column and the removal of muscle attachment points can result in excessive spinal movement and alterations in spinal biomechanics, leading to various postoperative complications [10]. Existing postoperative complications include the invasion of hematomas into the spinal canal, iatrogenic spinal stenosis and nerve root adhesion, spondylolisthesis, spinal instability, and deformities [5]. These complications may lead to persistent or recurring clinical symptoms post surgery, potentially necessitating further surgical interventions [6]. Additionally, a laminectomy can result in the proliferation of epidural fibrous tissue and scar tissue formation in areas of lamina defects (known as the laminectomy membrane), which can lead to new incidents of compression and make revision surgery more challenging [11]. Adults undergoing laminectomies for intradural lesion resections have been found to develop postoperative progressive spinal deformities in up to 20% of all cases [10,12]. Pediatric patients were at a greater risk of developing postoperative spinal deformities, with rates ranging from 20% and 100% [10,13]. Furthermore, the development of deformities often complicates functional outcomes and may necessitate additional fusion surgeries, contributing a poor prognosis in up to 50% of patients [14].

A recent trend in the surgical treatment of intraspinal lesions involves preserving the posterior elements to maintain their integrity and segmental stability, while also allowing for the adequate decompression of neural structures within the spinal canal [5]. Therefore, it has become commonplace to minimize damage to the original structure of the spine and perform anatomical and functional reconstructions [5]. In this pursuit, the laminoplasty, which aimed to restore the posterior elements, was initially introduced by Raimondi et al. in 1976 [15] as a solution to the limitations of the laminectomy. Subsequently, a range of surgical techniques and approaches for laminoplasties have been developed to address spinal occupying lesions or stenosis as an alternative to decompressive laminectomies [4,5,16]. Given that the use of miniplates has significantly advanced the restoration of posterior elements, this laminoplasty procedure has gained increasing popularity and serves as a standard posterior approach to the spinal canal for various surgeries, not limited to the cervical region but also extending to thoracic and lumbar lesions [4,5,16]. Currently, it is the most ideal surgical procedure for intraspinal pathologies in theory, with the exception of infections or malignant tumors of the bone [4,17].

Considering the overall satisfactory outcomes observed in laminar roof restorations with miniplates, Wiedemayer et al. [17] recommended the restorative laminoplasty technique as a valuable alternative over the laminectomy for a posterior approach in intraspinal surgery independent of age, the location of lesions, or the number of segments involved. As increased practice and experience with this technique grows, Wiedemayer et al. [17] stated that the restoration of posterior elements should no longer be used as a special technique for selected cases, but should be considered as a valuable alternative to the laminectomy as the standard posterior approach to spinal surgery. Although surgical methods in which the osteotomized laminar complex was repositioned rather than removed have long been performed without reported complications, the technique of using miniplates for restorative laminoplasties has not been widely accepted. Currently, there is insufficient convincing evidence to recommend this technique widely [4]. One possible reason for the limited clinical application of restorative laminoplasties could be the absence of internal fixation systems that are compatible with this technique, resulting in complex surgical procedures and inadequate levels of postoperative stability [4]. Additionally, the hesitation to use miniplates may stem from concerns regarding their mechanical strength, fixation efficacy, safety, and stability [4,5,18]. Therefore, in light of the potential risks involved, there is a danger they are ill-equipped when it comes to potential instrumentation issues and unintended consequences. These risks include issues like laminar fractures, laminar flap sinkage, the loosening of screws, and the failure of fixation devices, which often require additional surgery for correction [5,18]. In addition to other factors, the widespread use of pedicle screws was another significant factor influencing the widespread adoption of restorative laminoplasties [5]. Pedicle screws are currently the most commonly utilized internal fixation devices following laminectomies [5,19]. A laminectomy using pedicle screw fixation has increasingly become the preferred method for addressing this disorder, as it can help prevent postoperative instability and deformity, particularly in cases in which long segments need to be exposed [19]. Nevertheless, there continues to be ongoing debate in the literature regarding the necessity of posterior instrumentation for stabilizing the spine post laminectomy [19]. To date, there is no special internal fixation system that is suitable for the restorative laminoplasty technique. Orthopedic surgeons have primarily utilized miniplates from the metacarpals and phalanx or metatarsal miniplates in traumatic orthopedics, while neurosurgeons have used cranial miniplates for restorative laminoplasty procedures [4].

In recent years, surgeons have utilized L-shaped miniplates (LSMs) or two-hole miniplates (THMs) with preliminary shaping in restorative laminoplasties, yielding satisfactory outcomes [4,20,21,22,23,24,25]. Although these products are commonly used in clinical practice, the extensive shaping and bending required during surgery could damage the physiological and mechanical properties of the miniplates, prolong the surgical time, disrupt the surgical process, and potentially lead to unnecessary effects [4]. What is more serious is that excessive fatigue or even fractures could occur at the molding area, leading to severe complications [4]. To address these issues, H-shaped miniplates (HSMs) (Patent No. ZL 2014 2 0491485.4) that were specifically designed for restorative laminoplasties by our team were dedicated to restoring the posterior elements [4]. The design of HSMs was based on the physiological characteristics and anatomical structure of the lamina. The clinical experience of our institution with HSM fixation in restorative laminoplasties has shown their efficacy in treating intraspinal lesions.

Given the non-reproducible and unpredictable nature of clinical operations and therapeutic effects, along with the need for a further confirmation of the stability of restorative laminoplasties with the fixation of miniplates, it is necessary for us to determine the biomechanical properties of miniplates. In this regard, a biomechanical analysis can provide valuable and crucial insight into these conflicting observations. Our previous biomechanical research [4] has revealed that the mechanical strength of miniplates was sufficient to withstand body motion stress and spinal compression stiffness. Additionally, we found that HSMs exhibit a superior mechanical strength, segmental fixation effect, and levels of safety and stability when compared to LSMs and THMs [4]. It is also worth noting that the design and efficacy of new implants should be evaluated through biomechanical experiments using in vivo, in vitro, and in silico modeling to corroborate such postoperative alterations in spine kinematics and kinetics [26]. However, the biomechanical results in vivo and in vitro may vary due to differences in cadaveric specimens, animal samples, and patient characteristics [5]. In silico modeling investigations, by simulating changes in kinematics, offer an improved insight into postoperative alterations in the kinetics of segments [27]. The finite element (FE) model driven by pure moments with/without follower loads or driven by image-based displacements, as well as the musculoskeletal model with idealized passive joints, have been widely used to investigate the biomechanical behavior of implants and postoperative alterations of segments [28]. A force-controlled FE model failed to account for the crucial role of muscles, while a displacement-controlled FE model was sensitive to measurement errors in vertebral translations. It is widely recognized that the spine is influenced by various load conditions, among which the invariable rule is joint movement [29]. The movement of one segment corresponds to the mutual movement of another segment, which is a necessary prerequisite for understanding the interaction among an intact spine, a laminectomy, and a laminoplasty [30]. The goal of a restorative laminoplasty is to integrate the posterior column complex, and thus, the range of motion (ROM) and displacement of segmental vertebrae could directly reflect the biomechanical stability, efficacy, and feasibility of the surgery, as well as the stability of fixed segments and the risk of complications.

Notably, the literature is deficient in biomechanical comparisons of different miniplates for restorative laminoplasties, and no studies have attempted to clarify the biomechanical effects of restorative laminoplasties with the fixation of miniplates after laminectomies through an FE analysis [25]. There have been previous reports in the literature analyzing the biomechanical effects and instability of laminectomies using FE analyses [5]. Thus, the aim of this study was to determine whether a restorative laminoplasty with the fixation of miniplates, as a new surgical approach for intraspinal lesions, is more effective than a laminectomy in maintaining spinal motion, segmental movement displacement, and reducing postoperative instability through an FE analysis. This study also aimed to quantitatively analyze and compare the biomechanical efficacy and feasibility of various miniplates that share similar principles and functions. By evaluating the biomechanical variations in an intact spine, a spine after a laminectomy, and a spine after a restorative laminoplasty, we delved into the biomechanical properties of implanted miniplates after a laminectomy, with the goal of providing a theoretical basis for further clinical applications of restorative laminoplasties.

## 2. Material and Methods

### 2.1. Ethics Approval and Consent to Participate

This study was conducted in compliance with ethical standards and approved by the ethics committee (approval number: 81YY-KYLL-13-03). The spine imaging data were acquired through a human trial program following appropriate guidelines and regulations. Informed consent was obtained from the subject, who was fully briefed on the examination process and potential adverse effects and instructed to promptly report any reactions to the researchers. All testing was conducted on a voluntary basis, with confidentiality maintained and participant rights protected.

### 2.2. Research Object

The model was developed based on data collected from a single healthy adult volunteer to ensure the accuracy and reliability of the findings. A male volunteer (30 years old, 64 kg, and 176 cm) without history of chronic low back pain, tumors, trauma, fractures, deformities, or degenerative diseases involving the lumbar spine was recruited. Extensive radiological examinations of the lumbar spine and pelvis were conducted to confirm the absence or presence of any visible spinal damage or diseases.

### 2.3. Establishment of Intact Lumbar FE Model

A three-dimensional (3D), non-linear ligamentous attachment FE model of intact (INT) lumbar spine was adopted and modified for this research. Geometrical details were obtained from the volunteer in an unloaded neutral position by a high-resolution computed tomography (CT) scanner. The CT data set was supplied with the following technical parameters: voltage of 120 kV, current of 125 mA, scanning layer thickness of 0.625 mm, layer spacing of 0.625 mm, and spiral axis scanning from top to bottom. The image interval was 0.625 mm to maintain clearer image quality to facilitate measurement of lumbar endplate morphology.

A continuous series of images was exported and enlarged in order to identify different regions of tissues. A total of 403 CT scan layers of images were stored in DICOM format. The original DICOM data of normal vertebrae of the subject were imported into Mimics Research 19.0 software to establish a 3D model of lumbar spine. The noise points in point cloud image were reduced under the premise of ensuring the integrity of model. A threshold was set to differentiate bone and soft tissue, and soft tissue images around the bones were removed. The images were selectively segmented according to anatomical structure, and gray scale was adjusted appropriately to obtain a clear bone profile. Boolean calculation and interactive 3D manual/automatic cutting operations were performed to establish the rough 3D geometrical surface of the lumbar spine.

Following the completion of mask processing, the modified files of constructed vertebra were exported to STL format. These STL files were then repaired and optimized by Geomagic Studio 12.0 software to reconstruct solid surfaces by inversion. To smooth uneven surfaces caused by the stacking of CT images and form entity, the model was polished, filled, denoised, and solidified. Firstly, voids that should not exist on the surface of model were filled, non-characterized spikes and dents on the surface of model were removed, and the surface was smoothed and relaxed to prevent the occurrence of non-characteristic high-curvature and self-intersecting surfaces, so as to avoid unnecessary low-quality mesh division in the subsequent grid. Then, the encapsulation was completed by using the point cloud image, and triangular patch on the surface of model was reduced under premise of ensuring the shape of model so as to generate a continuous surface model (a solid model). Then, smooth nonuniform rational B-spline (NURBS) surfaces were implemented to fit discrete model surfaces to triangular patches, and finally, a 3D solid geometric model of the bony structure (L2/4) was reconstructed.

The rational non-uniform geometry structure was achieved, and subsequently, detailed processing was carried out to assemble each part of the structure into the INT model in SolidWorks 2022 software. Then, mesh was generated, and 3D model of surface mesh was further transformed into 3D model of volume mesh. This meshing process was performed for the fixation of miniplate systems together with lumbar spine itself to ensure the continuity of the mesh. The elements involved were created in geometric model according to anatomical structures of lumbar spine, including cortical bone, cancellous bone, endplate, annulus fibrosus, nucleus pulposus, articular cartilage, and ligament system. Finally, a complete 3D geometric solid model of L2/4 was formed, and subsequently, element setting, material property definition, biomechanical simulation, and analysis were carried out.

### 2.4. Defining Materials and Section Properties

A 3D FE model was developed to represent the intact L2/4, comprising three lumbar vertebrae, two intervertebral discs (IVDs), and associated ligaments. Detailed properties of all components in INT model are outlined in Table 1.

In this study, we assumed isotropic material properties for lumbar vertebrae, posterior elements, and IVDs. The developed L2–L4 segment model was meshed using C3D4 elements after a mesh convergence study. The cortical bone, cancellous bone, cartilage, ligaments, miniplates and screws, and facet were discretized according to the original geometry using C3D4 tetrahedron continuum elements. The cortical bone was defined as a 1.0 mm thickness outward from the outer layer of the cancellous bone based on CT image estimation. Difference between cortical and trabecular bone in posterior region was difficult to distinguish; therefore, the posterior elements were all assigned a single set of material characteristics which were different from that of the anterior structure. The starting and ending points of ligaments were set according to anatomical features, and their material properties were modeled as nonlinear tension-only connectors via a hypoelastic solid designation. To improve calculation accuracy, a hybrid mesh method was adopted to establish meshes of the model, and the different mesh types were established. This method allowed for more nodes and elements, which resulted in higher accuracy of biomechanical analysis.

The IVDs were modeled with annulus fibrosus, nucleus pulposus, and superior and inferior endplates. The discs were defined to be composed of 44% nucleus pulposus and 56% annulus fibrosus, and endplates were 0.5 mm thick. The heights of anterior and posterior edges of the IVDs were 9 mm and 7 mm, respectively. The superior and inferior boundaries of IVDs were assigned to the endplates of the adjacent vertebra, and outer boundaries of the IVDs were generated based on scanning geometry. The annulus was constructed as a circular structure between outer and inner annulus fibers. The annulus fibers were modeled with four layers of shell elements with a thickness of 1.5 mm. The annulus fibrosus was modeled as a composite of solid matrix with embedded fibers in the concentric rings around nucleus pulposus, and the fiber stiffness gradually increased from inside to outside. Each of the four concentric rings of ground substance contained two layers of evenly spaced circular fibers that were ±30° to the horizontal. The reinforcement structure annulus fibrosus was represented by truss elements with modified tension-only elasticity. In the radial direction, four double-cross-linked fiber layers were defined, and these fibers were bounded by annulus ground substance and both endplates. Both the incompressible and inviscid fluid-filled cavity of the nucleus pulposus and the hyperelastic properties of annulus matrix were modeled using the isotropic, incompressible, hyperelastic Mooney–Rivlin formulation. The nucleus pulposus was assumed to behave as a bag filled with incompressive fluid and was modeled by non-compressible solid tetrahedral linear elements within inner annulus fiber. Element members with low elastic modulus and large Poisson ratio were applied to simulate nucleus pulposus. The endplates were defined between two annulus fiber layers and modeled by solid tetrahedral elements. The thicknesses of endplates and facet cartilages were 0.8 mm and 0.2 mm, respectively. Curved surfaces of the healthy facet joints were carefully prepared and separated with a 0.60 mm gap in unloaded neutral position.

### 2.5. Boundary and Loading Conditions

The contact types were set according to previous studies [34,35,36,37,38]. A 3D surface-to-surface sliding contact with friction was specified to simulate the joints’ contact behavior with a finite sliding interaction defined to allow random motions, including sliding, rotation, and separation. Articulating friction was neglected, and only transmitted normal forces were considered. In the interaction settings, cortical bone was bound to cancellous bone, and ligaments were bound to the outer surface of cortical bone. The contact between facet joints was defined as nonlinear, surface-to-surface, frictionless sliding contact [34,38]. The contact between vertebra and IVDs was designated as mutual contact with a friction coefficient of 0.08 [37]. The contact type of miniplate and lamina was set as friction with a friction coefficient of 0.2 [35]. The contact type of screw and lamina was set to binding mode [35,36].

The boundary and loading conditions used during the analysis were derived from Yamamoto’s in vitro studies [39]. For all FE models, all directions (flexion, extension, left and right lateral bending, and left and right axial rotation) at the inferior surface of L4 were fully constrained, with no displacement or rotation allowed. A reference point was established at the center of upper surface of L2 and coupled with the upper surface. A vertical axial preload of 500 N, representing upper body weight and muscle forces, was applied to the reference point of L2. A torque of 10 Nm, which could cause the lumbar spine to move within the physiological range without destroying its organizational structure, was applied to the reference point of L2. The forces were jointly simulated, and ROM and displacement of the relevant segment under physiological conditions were simulated under the six directions (Figure 1). The ROM and displacement of L2/3, L3/4, and L2/4 were recorded and compared with Yamamoto’s in vitro studies [39] to validate the model.

### 2.6. The Mesh Sensitivity and Convergence Test

To verify the accuracy of aforementioned FE model, mesh sensitivity analysis and convergence test were performed on INT model. The geometry of tissues was meshed with tetrahedral elements using SolidWorks 2022 software, and an automatic algorithm was used to generate 10-node tetrahedral solid elements to mesh the various solid models. The quality of all elements was monitored using aspect ratio and Jacobian check to avoid obvious discontinuities and unrealistically high stress concentrations. A mesh convergence test was conducted to confirm that the predictions of INT model were not significantly influenced by mesh resolution. Different mesh sizes may have different results. The results of smaller mesh were relatively reliable; however, smaller mesh was more cumbersome and time-consuming.

To find a suitable mesh size for this test, the seed densities were gradually decreased for meshes in INT models, including Mesh 3.0 mm, Mesh 2.5 mm, Mesh 2.0 mm, and Mesh 1.0 mm. Under the same boundary and loading conditions, the displacement of the reference point at the center of L2 superior endplate was measured under a compressive preload of 500.0 N combined with a pure moment of 10.0 Nm. The convergence of INT model was verified by analyzing displacement of reference point in the four mesh sizes models. Mesh refinement was executed for modeling accuracy until excellent monotonic convergence behavior with less than 5% difference between the predictions obtained by two successive mesh resolutions in the L2-top displacement was achieved.

The elements and node numbers for different mesh resolutions were presented in Table 2. The resolution difference between predicted results of the 2.0 mm and 1.0 mm mesh sizes was less than 5% in most tissues. This indicates that the mesh size between 2.0 mm and 1.0 mm was considered optimal for our study. Therefore, the 2.0 mm mesh size could be considered convergent, while the 1.0 mm mesh size was not used, as it would add computational costs with a negligible difference in the results. The mesh size was adjusted to 2.0 mm to improve accuracy of the simulation results and the efficiency of calculation. The convergence tests were also performed on miniplates and screws. During surgery, miniplates were often bent to ensure proper fixation; therefore, the mesh of the miniplates was deformed to accommodate placement of screws on laminar side. The seed densities were gradually decreased in miniplates and screws for meshes, including Mesh 1.5 mm, Mesh 1.25 mm, Mesh 1.0 mm, and Mesh 0.75 mm. The resolution difference between the predicted results of the 1.0 mm and 0.75 mm mesh sizes was less than 5%. Thus, the mesh size of miniplates and screws was adjusted to 1.0 mm. The final number of elements and nodes used in this study is presented in Table 3.

### 2.7. Validation and Modification of INT Model

Indirect validation studies aimed to extensively verify the accuracy of aforementioned intact FE model using automated algorithms by contrasting results from previous models or samples with experimental data in the literature [39]. Experimental and numerical comparisons were used to validate the simplifications and assumptions of FE model. In this study, we reconstructed the completed model based on the original model and subsequently compared it with the in vitro biomechanical findings reported by Yamamoto et al. [39], focusing on ROM. The newly reconstructed intact FE model was tested and validated under the six directions for the same loading conditions. The ROMs at L2/3 and L3/4 in INT model were the primary parameters chosen for validation.

### 2.8. Construction of Surgical FE Model

The laminectomy and restorative laminoplasty models were simulated and developed based on conventional surgical protocols (Figure 2). According to the needs of our research, the INT model of L2/4 has been slightly modified, which did not affect the simulation degree of operation models.

To simulate the laminectomy model, the medial parts of lamina, including spinous process, were osteotomized on both sides at their bases in the model of intact spine. Additionally, the associated ligaments (SSL, ISL, and LF) were also resected to accurately replicate the surgical procedure, while the facet joints remained intact.

To simulate the restorative laminoplasty models, the lamina and PLC were repositioned over the original site. After completion of laminectomy and intraspinal surgery at L2/4 lamina, a bicortical fracture gap of 2 mm in width at the junction of laminas was simulated by completely removing a layer of elements. Screw holes were created based on the desired plate position. The miniplates and screws were inserted into laminectomy model. Computer-aided design models of miniplates and screws were generated from the physical dimensions. Due to the limited space within lamina, bifurcation of implanted screws prevents collision among screws or lamina fractures. Thus, three fixations of different miniplates in restorative laminoplasty models were reconstructed. Figure 3 illustrates the miniplates and models used in this study.

### 2.9. Statistical Analysis

A calculation was performed after successfully building the models under settled boundary and loading conditions. The ROM and URES distributions were calculated for INT model, LE model, RL-THM model, RL-LSM model, and RL-HSM model. URES is the resultant displacement without using reference geometry, and maximum URES was noted by recording the maximum resultant displacement.

For statistical analysis, the calculated results were replicated for mesh convergence tests to reduce potential errors caused by mesh size. With altered mesh size in a four-time technical replication (Table 2), results that showed an almost stable solution with a variability of less than 5% were recognized as acceptable and recorded in this study [27]. Descriptive statistics were used to present the trend derived from comparison and to provide direct tables for analysis.

Compared with INT model, the percentage increment (% increment) in ROM or maximum URES was calculated using the following equation.
% increment = (|ROM/URES _INT_ − ROM/URES _surgery_|/ROM/URES _INT_) × 100

Similarly, compared with laminectomy model, the percentage reduction (% reduction) in ROM or maximum URES was calculated using the following equation.
% reduction = (|ROM/URES _laminectomy_ − ROM/URES _laminoplasty_ |/ROM/URES _laminectomy_) × 100

Furthermore, in comparison with the RL-HSM model or RL-LSM model, the percentage changes (% changes) in ROM or maximum URES were calculated using the following equation.
% changes _RL-HSMs_ = (|ROM/URES _RL-HSMs_ − ROM/URES _RL-LSMs or RL-THMs_|/ROM/URES _RL-HSMs_) × 100
% changes _RL-LSMs_ = (|ROM/URES _RL-LSMs_ − ROM/URES _RL-THMs_|/ROM/URES _RL-LSMs_) × 100

The percentage variation in different procedures was compared, and a difference of more than 5% was considered significant [28].

## 3. Results

### 3.1. Validation of the INT Model

To validate the INT model, kinematics data from the present INT model were compared with in vitro experimental data obtained from cadavers as reported by Yamamoto et al. [39]. The reference of Yamamoto et al. [39] is overly used in FE analyses [2,3,4,5,6,7,8,9,10,11,12,13]. The ROM of the intact L2/4 model under six directions was tested and validated, using the results from in vitro cadaveric tests for the same loading conditions (Table 4). At L2/3, the differences between the INT model and the literature data were found to be within 3.08% in flexion, 2.79% in extension, 0.71% in left lateral bending, 0.86% in right lateral bending, 3.18% in left axial rotation, and 3.67% in right axial rotation. Similarly, at L3/4, the differences were within 3.20% in flexion, 2.97% in extension, 2.07% in left lateral bending, 2.81% in right lateral bending, 0.74% in left axial rotation, and 0.80% in right axial rotation. A slight discrepancy in the outcomes between the present model and the reference study indicated that the actual material and material properties of the cadaveric and numerical analyses may have differed from those of the present study. However, it was highlighted that [27], despite the unavoidable variation in results, the material properties used in the present study accurately capture the experimentally and numerically observed trend in the results. All segmental ROM results from the INT model were within the effective range or average standard deviation of the cadaveric experiments of Yamamoto et al. [39] (with a <5% difference). Furthermore, the results from the present FE models also showed the same variation trends as the study conducted by Yamamoto et al. [39], indicating that the intact L2/4 FE model in this study was successfully constructed and could be used for further biomechanical analyses. Since all the data were conformed through normal human body parameters, the INT model was able to replicate human physiological movements of the L2/4 vertebrae. Owing to modeling inconsistencies, some individual values were inconsistent with previous FE results [27,28,30]; however, despite these variations, all values under different conditions were within the vitro range.

### 3.2. The Quantitative Values of ROM

The ROM data are expressed as an angle, which directly reflects the stability of the relevant segments. The lower the ROM value, the more stable the relevant segments and the lower risk of complications. Figure 4 presents the ROM values at L2/3, L3/4, and L2/4 in all of the models. When considering the quantitative values of segmental and overall ROM for all directions, the LE model exhibited the highest ROM, followed by the RL-THM model, the RL-LSM model, the RL-HSM model, and the INT model.

### 3.3. Comparison of ROM Variation between the INT Model and Different Surgical Models

Table 5 and Figure 4 demonstrate that the INT model experienced the greatest ROM increment for all surgical segments (L2/3, L3/4, and L2/4), followed by the RL-THM model and the RL-LSM model, with the smallest increment observed in the RL-HSM model. After the laminectomy, the largest increase in ROM for all surgical segments was observed under axial rotation, followed by extension and flexion, with the smallest increase seen in lateral bending. Following the restoration of posterior elements with three miniplates, the most significant increase in ROM for all surgical segments was observed under axial rotation. In the RL-HSM model, the ROM results for all surgical segments under lateral bending and the ROM results at L3/4 under flexion and extension were similar to those in the INT model (a <5% difference). The ROM increments at L2/3 and L2/4 were within 7.42% and 5.90% under flexion and 6.51% and 5.22% under extension. In the RL-LSM model, the ROM results at L3/4 under flexion and right lateral bending were comparable to those in the INT model (a <5% difference), while the remaining ROM results were higher than those in the INT model (a >5% difference). In the RL-THM model, the ROM results for all surgical segments were higher than those in the INT model (>5% difference).

### 3.4. Comparison of ROM Variation between LE Model and Different Restorative Laminoplasty Models

Table 6 and Figure 4 demonstrate that the largest reduction in ROM under the six directions from the LE model to the matched restorative laminoplasty models for all of the surgical segments was noted in the RL-HSM model, followed by the RL-LSM model and the RL-THM model. This reduction was most pronounced for axial rotation, followed by extension, flexion, and lateral bending. Interestingly, the ROM reduction under flexion at L2/3 in the RL-THM model was slightly greater than that under extension (a <5% difference), contrary to the general trend observed.

### 3.5. Comparison of ROM Variation among Different Restorative Laminoplasty Models

Table 7 and Figure 4 demonstrate that there were no significant differences in ROM values among the three types of restorative laminoplasty models under flexion (<5% difference). Compared to the RL-HSM model, both the RL-THM and RL-LSM models showed an increase in the quantitative values of ROM under six directions for all of the surgical segments. The most notable increase was observed in axial rotation (a >5% difference). Notably, there were no significant differences in ROM between RL-HSMs and RL-LSMs for all of the surgical segments under lateral bending (a <5% difference) and no significant changes in ROM at L3/4 and L2/4 under extension (a <5% difference). Additionally, there were no significant changes in ROM between RL-HSMs and RL-THMs under extension and lateral bending at L3/4 (a <5% difference), but significant differences were observed at L2/3 and L2/4 (a >5% difference). Furthermore, when comparing the RL-LSM and RL-THM models, the latter showed increased ROM values under six directions for all of the surgical segments, with the most significant increase observed in axial rotation. However, there were no significant differences in ROM values between RL-LSMs and RL-THMs for all of the surgical segments under flexion, extension, and lateral bending (<a 5% difference).

### 3.6. The Maximum URES Values and Their Variations in All FE Models

Table 8 and Figure 4 and Figure 5 present the URES distribution under the six directions. For the quantitative value of the maximum URES, the LE model yielded the highest URES values under the six directions, followed by the RL-THM model, the RL-LSM model, the RL-HSM model, and the INT model.

Compared to the intact state, a significant increment (a >5% difference) in the normalized URES values was observed following the laminectomy and restorative laminoplasty under the six directions. Among the four surgical models, the LE model exhibited the highest URES increment, followed by the RL-THM model, the RL-LSM model, and the RL-HSM model. Across all laminectomy types, axial rotation resulted in the greatest URES increment compared to that of the INT model, followed by extension, flexion, and lateral bending. The URES differences between the LE model and INT model were within 109.97% under axial rotation, 69.01% under extension, 48.83% under flexion, and 31.20% under lateral bending. Similarly, for all laminoplasty types, the axial rotation produced the highest URES increment compared to that of the INT model, followed by flexion, extension, and lateral bending.

The restorative laminoplasty with the fixation of miniplates demonstrated a significant decrease in normalized URES values compared to those of the LE model under the six directions (a >5% difference). The RL-HSM model showed the greatest reduction in URES values, followed by the RL-LSM model and then the RL-THM model. Across all types of restorative laminoplasty, axial rotation produced the highest reduction in URES values compared to those of the LE model, followed by extension. The reduction in URES under flexion was similar to that under lateral bending for all types of restorative laminoplasty (a <5% difference).

A notable increase in URES from the RL-LSM model and the RL-THM model to the matched RL-HSM model was noted, particularly under axial rotation and flexion. The differences in URES between the RL-LSM model and the RL-HSM model were insignificant under extension and lateral bending (a <5% difference). However, the RL-THM model exhibited a significantly higher URES in comparison to that of the RL-HSM model under extension and lateral bending (a >5% difference).

In comparison with the RL-LSM model, the maximum URES variation in the RL-THM model was found to be within 11.27% under axial rotation, followed by 7.65% under lateral bending, and 7.42% under flexion. Interestingly, there was no significant change was observed in the RL-THM model under extension, with an increase of only 2.75%.

## 4. Discussion

Laminectomies have long been regarded as a common surgical intervention to address intraspinal occupying lesions, lumbar degenerative diseases, and spinal stenosis [4,5]. The safe resection of intraspinal lesions depends on the adequate exposure of the spinal canal and its surrounding structures, which requires surgeons to remove part or even all of the posterior elements to obtain a sufficient level of visibility and operating space [4]. In surgical terms, the resection margin increases from unilateral to bilateral interlaminar decompression and reaches the greatest extent for laminectomies [5]. A conventional laminectomy disrupts the normal bony attachment points of paravertebral muscles and impairs the function of posterior elements as a tension band [5]. As a consequence, after a laminectomy, the spine structure may be vulnerable to segmental instability at the surgical segments, particularly in pediatric patients, those undergoing cervical spine surgeries, and those with pathologies involving multiple segments [11]. To date, there remains no clear consensus on the determination and definition of a possible instability or possible risk factors for its development [16]. Although a laminectomy does not lead significantly to spinal instability, it is unable to prevent the hypermobility of surgical segments, which could result in the progression of kyphotic deformities and spondylolisthesis in the long term due to the disruption of biomechanical balance [40]. The laminectomy membrane, which is epidural scar tissue in the spinal canal, could result in negative outcomes following the removal of posterior elements [41]. Additionally, iatrogenic instability may be underreported secondary to its relationship to segmental surgery and surgeon bias [28].

To address potential issues of laminectomies, scholars have proposed various approaches for several conditions, including radical laminotomies, facet-sparing laminectomies, osteoplastic laminotomies, and pedicle screw fixation [42,43]. In the latter half of the 20th century, radical laminotomies were a commonly performed surgical procedure for treating intraspinal diseases [44]. This procedure involved removing not only the lamina but also the pars and facets. Although radical laminotomies have been proved to be effective in relieving neuro-compressive symptoms, at least in the short-term, postoperative instability has been frequently noted afterwards [44]. In 1990, Abumi et al. [45] evaluated the stability of cadaveric lumbar spines following facet preservation. Their findings indicated that a facet-sparing laminectomy yielded a stable spine, and a complete laminectomy, whether unilateral or bilateral, but was not recommended as it also predisposed the spine to instability. Despite this, in clinical practice, when surgery is required, facet-sparing laminectomies remain the most commonly performed treatment for decompressing the spinal canal and associated nerve roots [44]. During a facet-sparing laminectomy procedure, it is advised to retain a minimum of 50% facet bilaterally and sufficient pars to avoid spine instability [46]. However, even with these precautions, a single-level facet-sparing laminectomy might result in the advancement of the curve or the progressive symptomatic postoperative instability of a scoliotic lumbar spine. Studies have reported the incidence of iatrogenic spondylolisthesis post facet-sparing laminectomy to range from 8% to 31% [46,47].

To alleviate the symptoms stemming from facet-sparing laminectomies, procedures like partial or hemi-laminectomies, such as osteoplastic laminotomies, not only decompress the spinal canal but also help preserve PLC and midline structures, thereby achieving spinal stability and clinically effectiveness by aiding in stabilizing spinal movement [43]. The osteoplastic laminotomy procedure was designed to protect the spinous process, interspinous ligaments, and facet capsules, removing only the bilateral half of the inferior part of the upper lamina and a limited amount of the superior part of the lower lamina, including the adjacent ligamentum flavum in the treated segments. An in vitro study compared the biomechanical potential for the instability of the lumbar spine following a facetectomy and osteoplastic laminectomy [48]. The results demonstrated that the ROM in axial rotation after the facetectomy remarkably increased by 113% over the intact state, but only increased by 57% after the osteoplastic laminectomy. Additionally, the ROM in flexion/extension increased by 33% after the facetectomy, with no significant change observed after the osteoplastic laminectomy. Furthermore, there was no significant increase in ROM for lateral bending following either procedure. Zander et al. [49] found that a unilateral hemi-facetectomy increased segmental rotation for the loading situation of torsion. Expanding the resection to a bilateral hemi-facetectomy increased the segmental rotation even more, while a further resection up to a bilateral laminectomy had only a minor additional effect. In terms of axial rotation, spinal stability was decreased even after a hemi-facetectomy or hemi-laminectomy. The hemi-laminectomy and radical laminotomy only differed in their effect on muscle-supported flexion/extension and lateral bending. The study determined that the osteoplastic laminectomy maintained a greater level of spinal stability than that of the facetectomy and that patients should therefore avoid excessive axial rotation after such treatments. Both the facet-sparing laminectomy and osteoplastic laminotomy could better reduce biomechanical damage, but on the other hand, they also posed a higher risk of spinal cord injury due to limited visibility during surgery. Previous in vitro studies have demonstrated that these techniques could reduce the threshold at which shear forces and torsion moments cause lumbar failure [50,51,52]. Such effects are likely smaller than the biomechanical changes caused by more extensive or multilevel decompressive surgeries [53,54]. However, decompression techniques like radical laminotomies, facet-sparing laminectomies, and osteoplastic laminotomies, which involve invasive procedures on the posterior column, may have the potential to compromise spinal stability and lead to kyphotic instability. Subsequently, after more than 100 years of improvements in laminectomies, the internal fixation system was added.

As a countermeasure against kyphosis, additional posterior interbody fusion with pedicle screws was recommended to enhance spine stability and decrease the incidence of postoperative kyphosis, forming the current laminectomy procedure with pedicle screw fixation surgery [27]. Serving as the main force for the multi-plane stability reconstruction of lumbar vertebra, this technique has provided a relatively wide exposure of the spinal cord and can be easily extended in either the rostral or caudal directions. As a result, it has gradually become the preferred surgical approach for this condition, particularly in cases in which extensive spinal exposure is needed [27,55]. Pedicle screw devices are aimed primarily at arthrodesis (fusion), which could be used to form a rigid construct with the spine to replace bone, restore alignment, maintain position, and prevent movement for managing various spinal instabilities, deformities, and painful conditions. The instrumented fusion with pedicle screws was indispensable for permanent stabilization, but it also brought some unique complications and disadvantages [56,57]. In terms of motion, pedicular fixation with instrumented fusion restricted segmental and overall spinal mobility between vertebrae, potentially causing a redistribution of loads to adjacent segments and increasing the risk of premature degeneration in those areas [57].

In an attempt to minimize the problems of fusion-related morbidity, the concept of the dynamic reconstruction of the spine was introduced in 2002 [58]. This approach was intended to be performed with a non-rigid but stable non-fusion device. Since then, numerous non-fusion posterior dynamic stabilization systems (DYSSs) have been introduced to the market in recent years [55]. Biomechanically, the design of DYSSs was intended to permit axial load distribution. In vitro experiments have demonstrated that DYSSs offer substantial stability and could therefore be considered as an alternative to fusion surgery in these indications while preserving the motion segment. The DYSSs were found to preserve motion segments, and the lowest motion-restricting effect was that for axial rotation, no matter if an intact or a destabilized segment was instrumented [59,60]. Panjabi et al. [60] concluded that DYSSs effectively stabilized the decompressed spinal level in sagittal and frontal planes without producing significant adjacent-level effects compared to fusion. Additionally, Schmoelz et al. [61] found that DYSSs restricted motion in intact spinal motion in flexion/extension and lateral bending, while providing a limited stabilizing effect in axial rotation. Although the DYSSs retained some motion and reduced the load on adjacent levels compared to rigid fixation, they only retained limited motion and may potentially stress the adjacent level above. Therefore, close monitoring for adjacent segment disease was necessary, especially in cases of multiple-level dynamic stabilization. In a long-term follow-up study, only 8% of patients maintained the motion with DYSSs, while adjacent segment degeneration occurred in up to 47% of cases [61]. A key factor for achieving a favorable clinical outcome seemed to be the use of an implant, which offers adequate levels of stability while permitting limited motion. As the clinical use of pedicle screws has increase and more in-depth studies have been conducted, it is evident that postoperative issues, such as loosening and breakage, are becoming more common and challenging. Previous research has indicated that screw breakage commonly occurs around the thread shank region, with an incidence ranging from 2.6% to 60% [55]. Various efforts have been made over the years to enhance the stability of the screw–bone interface and improve posterior fixation strength [62]. Despite advancements in the design, material, and motorization of pedicle screws and connecting rods, reports of breakage remain relatively common in both fusion and non-fusion devices [62]. Moreover, fusion and non-fusion devices face similar limitations in clinical practice due to the drawbacks of laminectomies [63]. Consequently, the choice of performing laminectomies with pedicle screw fixation has sparked debates among scholars regarding the necessity and effectiveness of fusion versus non-fusion surgery in decompressive procedures [63].

Recently, the adequate exposure of the spinal canal, complete removal of intraspinal lesions, sufficient release of spinal cord compression, effective maintenance of spinal integrity, and utmost protection of spinal anatomy and function have gradually become the basic principles of intraspinal surgery [4]. The restoration of the posterior elements while maintaining respect for normal anatomy is considered crucial in intraspinal surgery. The laminoplasty, initially proposed by Raimondi et al. [15] in 1976, has emerged as an alternative to the decompressive laminectomy due to its advantages. Various options such as the double-door laminoplasty by Kurokawa et al. [64] in 1982 and the open-door laminoplasty by Hirabayashi et al. [65] in 1983 have been reported and advocated. The term “laminoplasty” typically involves creating space through a laminectomy without removing the lamina. However, some confusion exists regarding the correct term to be used for a laminoplasty [66,67]. Currently, the mainstream laminoplasty techniques can be broadly categorized into two categories: the enlargement of spinal canal, represented by expansive laminoplasties, like open-door and double-door laminoplasties; and the restoration and reconstruction of the spinal canal, represented by restorative laminoplasties. The primary goal of a laminoplasty is to reposition the lamina to expand or restore the spinal canal, allowing the spinal cord to migrate posteriorly. Therefore, depending on the type of surgery, we suggest that restorative laminoplasties and expansive laminoplasties are more appropriate for this term and more in line with its technical features.

Ever since the original expansive laminoplasty outlined by Hirabayashi et al. [65] and Kurokawa et al. [64], various technologies have been developed to assist in the procedure and are now widely used in clinical practice [4,5,6,18,22,68]. The use of ultrasonic osteotome devices and miniplates has significantly improved the enlargement and restoration of posterior elements, making this procedure increasingly popular as a standard posterior approach for different intraspinal surgeries in the cervical region [5,6]. Biomechanical testing and FE analyses have demonstrated that expansive laminoplasties with the fixation of miniplates could enhance spine stability, compressive resistance, and resistance to bending, shear, and rotation [28,69]. Expansive laminoplasty techniques have been extensively utilized in clinical practice, with specific miniplates designed for these procedures including Centerpiece, Neulen, Arch, and Leverage miniplates [5,6]. These miniplates are well established products with proven efficacy in clinical settings. Nevertheless, there is currently no internal fixation system that fully aligns with the principles of restorative laminoplasties.

Despite being introduced nearly 5 decades ago, the restorative laminoplasty is not used as a posterior approach to spinal surgery; only a very small number of institutions have adopted this technique for intraspinal surgeries [4,20,21,22,23,24,25]. The purported advantage of the restorative laminoplasty was the structural recovery of posterior elements, which theoretically return the spine to more normal biomechanics. It was considered a practical method for restoring normal anatomical structures, maintaining postoperative stability, and preventing kyphotic deformities when compared to a laminectomy [4,20,21,22,23,24,25]. A restorative laminoplasty allows for wider intradural access and can be used as an approach to the spinal canal when the bone decompression of the vertebral canal is not the primary goal of the surgical procedure. Additionally, a restorative laminoplasty for intraspinal surgery enables surgeons to prevent postoperative bleeding and the invasion of hematoma and scar tissue and facilitates uncomplicated access for any subsequent surgeries. It has been demonstrated that the restorative laminoplasty is a versatile surgical option that is not limited by age, surgical site, or the number of affected segments [19]. There is little debate that a laminoplasty is a more practical surgical technique than a laminectomy for the treatment of intraspinal pathologies [19]. In this case, a restorative laminoplasty should be a potentially valuable option; however, it has not yet gained widespread recognition or application [4,20,21,22,23,24,25]. The use of miniplates for restoring posterior elements has predominantly been limited to specific certain cases, such as cervical surgeries or pediatric cases, with limited research on its application in the thoracolumbar spine [4,20,21,22,23,24,25]. One of the reasons for the limited utilization of the restorative laminoplasty may be attributed to the absence of randomized studies showcasing its superiority over the laminectomy. The available evidence in favor of the restorative laminoplasty is mainly at level III. Additionally, the initial technique for restorative laminoplasties lacks suitable internal fixation systems in clinical settings. Clinicians have expressed concerns regarding the strength and fixation efficacy of miniplates and screws, particularly in the thoracolumbar spine, where stress loads are significant. There were apprehensions about the ability of miniplates and screws to hold the lamina in position until solid bone fusion was achieved, as well as the risk of loosening and breakage due to torsion, potentially leading to posterior fixation failure. Orthopedic and neurosurgical surgeons currently rely primarily on conventional bone plates that are manufactured in standard shapes and sizes [4,20,21,22,23,24,25]. These plates can be immediately used in emergency surgeries and can be produced in a cost-effective manner. Existing bone plates for other body regions have been accepted as a solution with mostly satisfactory results [4,20,21,22,23,24,25]. However, their long-term safety and stability remain unknown and require further confirmation.

Several commercially available miniplates have been utilized for restorative laminoplasties; nevertheless, these miniplates are primarily designed for use in the hand, foot, or skull, each with distinct mechanical profiles, morphological differences, and structural variations [4,20,21,22,23,24,25]. Unfortunately, these plates were not specifically tailored for restorative laminoplasties and do not perfectly align with individual laminar anatomy. For each case and use, they require extensive shaping and bending during surgery to achieve a proper fit [4,20,21,22,23,24,25]. Despite these modifications, there is still a risk of biomechanical or anatomical mismatch, which could lead to stress concentration, excessive fatigue, and an increased risk of plate or screw failure, bone malunions, or other serious complications. In orthopedic or neurosurgical surgery and in the field of bioengineering, there is a growing interest in the utilization of subject-specific plates for bone fixation. This is especially important when conventional bone plates do not align accurately with an individual’s unique anatomy [26]. Therefore, HSMs based on the physiological characteristics and anatomical structure of the lamina were specially designed for restorative laminoplasties. These HSMs have been clinically proven to be effective in the restoration of posterior elements of the spine and the treatment of intraspinal lesions at our institution. Currently, the mainstream miniplates for restorative laminoplasties are categorized as linear-shaped miniplates (e.g., THMs) [23], triangular-shaped miniplates (e.g., LSMs) [22], and quadrangular-shaped miniplates (e.g., HSMs) [4].

Despite the use of the fixation of miniplates in restorative laminoplasties, which has been considered the most popular technique in expansive laminoplasties, there are insufficient data available regarding their clinical and radiologic outcomes, as well as limitations [4,20,21,22,23,24,25]. The current available literature regarding restorative laminoplasties is primarily composed of clinical research or technical notes, with assessments based on subjective scoring and measurements of sagittal angles using medical imaging [4,20,21,22,23,24,25]. Clinical practitioners have also considered the advantage of modified procedures which lack the support of basic research [4,20,21,22,23,24,25]. Additionally, there is a dearth of comparative studies on restorative laminoplasty techniques, leading to ongoing debate regarding the optimal approach for treating intraspinal pathology. Uncertainty remains regarding whether these procedures effectively reduce the motion and displacement of relevant segments, as well as the associated risks of spinal instability and delayed deformities. Anatomical and biomechanical studies investigating this advanced technique are notably absent. Currently, there are no published reports analyzing the biomechanical effects and instability resulting from motion and displacement in restorative laminoplasties with miniplate fixation using FE analyses. Furthermore, the existing literature lacks comparative studies among different types of miniplates. To achieve stable bone fixation, satisfactory bone union, and complete functional results, it is essential to consider the biomechanical requirements during plate design and manufacturing [26]. One of the top concerns in using miniplates for restorative laminoplasties is their mechanical strength in holding laminar bone. Our previous research has shown that the mechanical strength of HSMs was adequate to withstand the stress of body motion [4]. This research primarily focused on assessing the stability of motion segments following a restorative laminoplasty post laminectomy. An FE model of L2/4 was developed to quantify the biomechanical effects of the restorative laminoplasty with miniplate fixation on motion and displacement during flexion, extension, lateral bending, and axial rotation under maximal loading conditions.

From a holistic perspective, the increase in motion and displacement under the six directions was most pronounced in the laminectomy compared to the intact state, followed by RL-THMs, RL-LSMs, and RL-HSMs, respectively. The investigated miniplates showed a motion-restricting effect on laminectomy spinal motion systems in terms of ROM and URES. The greatest reduction was observed in HSMs, followed by LSMs and THMs, respectively, indicating that HSMs were more effective than LSMs and THMs in maintaining motion segment stability.

Compared to the intact state, the laminectomy state showed the highest increase in motion and displacement. The largest increases in ROM and URES were observed during axial rotation, followed by extension, flexion, and lateral bending. Following the laminectomy, the ROM at the surgical segments (L2/3, L3/4 and L2/4) increased by 67.82%, 52.22%, and 60.02% in axial rotation; 36.51%, 27.89%, and 32.61% in extension; 29.55%, 19.03%, and 23.97% in flexion; and 19.56%, 15.78%, and 17.81% in lateral bending, respectively. The maximum differences in URES between the laminectomy model and intact model were 109.97% for axial rotation, 69.01% for extension, 48.83% for flexion, and 31.20% for lateral bending. It is important to note that ROM and URES are distinct concepts, with differing levels of variations. Despite these differences, the overall numerical trend remains consistent. These findings align with previous in vitro results. Quint et al. [70] discovered a substantial increase in ROM during a laminectomy under submaximal loading conditions of 7.5 Nm. They observed a 117.4% increase in axial rotation, 35.0% in extension, 32.0% in flexion, and 14.3% in lateral bending. Their study concluded that the ROM values were significantly reduced in the laminectomy state during flexion/extension and axial rotation, while the ROM values under lateral bending were stabilized. Schmoelz et al. [61] conducted a study on fresh human cadaveric lumbar spine (FHCLS) specimens (7.5 Nm) post laminectomy, revealing that the laminectomies significantly increased ROM values in both treated segments (L4/5 and L3/4) to 187% and 234% of the intact specimen under axial rotation. Furthermore, under flexion/extension, the laminectomies led to a 135% increase in ROM in both segments, while under lateral bending, it resulted in approximately 125% of the intact state. These findings underscore the clinical importance of stabilization following bilateral decompression with laminectomies. In the in vitro study (10 Nm) conducted by Strauss et al. [71], the ROM value at L4/5 increased by 31.12% under axial rotation, 23.64% under flexion/extension, and 2.62% under lateral bending. Their study concluded that the laminectomy displaced the balance point ventrally, and a ligamentoplasty restored it to a dorsal position, with the least impact observed in rotation. The instability observed post laminectomy in our present study was also consistent with findings reported by Fuchs et al. [72], Hamasaki et al. [73], Grunert et al. [74], and Smith et al. [75] in their in vitro studies, as well as those by Zander et al. [49], Bresnahan et al. [76], and Liu et al. [77] in their FE studies. These results of FHCLS were important, because our FE study demonstrated that a laminectomy resulted in the highest degree of instability with increases of 60.02% and 109.97% for the motion and displacement, respectively. Additionally, the lowest effect of the restorative laminoplasty was also found under rotation in the RL-HSM model for motion and displacement, respectively.

When comparing the effects of the restorative laminoplasty to the intact state, it was found that performing the restorative laminoplasty after a laminectomy with HSM fixation led to increases of 11.75%, 10.98%, and 11.37% in ROM under axial rotation at the treated segments (L2/3, L3/4, and L2/4). The increase in ROM for all of the treated segments was less than 5.00% under lateral bending. Additionally, the ROM increased by 7.06% and 5.66% under flexion/extension at L2/3 and L2/4, with the increase in ROM being less than 5% under flexion/extension at L2/4. In the RL-LSM state, the ROM increased by 21.19%, 18.11%, and 19.65% under axial rotation; 6.92%, 5.34%, and 6.19% under lateral bending; and 10.55%, 5.18%, and 7.85% under flexion/extension. In the RL-THM state, the ROM increased by 33.16%, 26.20%, and 29.68% under axial rotation; 10.64%, 9.60%, and 10.16% under lateral bending; and 13.30%, 6.45%, and 9.86% under flexion/extension. Furthermore, across all types of restorative laminoplasties, axial rotation resulted in the highest displacement increment compared to that of the intact model, followed by flexion, extension, and lateral bending. These findings indicated that the restorative laminoplasty was effective in preserving lumbar stability when compared to the intact state. This efficacy was particularly evident in lateral bending and flexion/extension, with the least impact seen in axial rotation. However, the issue of the limited recovery of axial rotation function is a common challenge in laminectomy treatments [54,78]. Even after procedures such as facet-sparing laminectomies, osteoplastic laminotomies, and expansive laminoplasties, the axial rotation function may still be diminished. These procedures were found to be significant in improving flexion/extension and lateral bending but were less effective in improving axial rotation. Consequently, patients undergoing restorative laminoplasties should be cautious of excessive axial rotation.

In comparing a restorative laminoplasty to a laminectomy, it was observed that the motion and displacement resulting from the laminectomy decreased gradually with the restoration of the posterior structures. The restorative laminoplasty with the fixation of miniplates was found to restrict the movement and displacement caused by the laminectomy, especially in axial rotation, followed by extension, flexion, and lateral bending. Among three restorative laminoplasty models, the RL-HSM model exhibited the greatest reductions in ROM and URES values, followed by the RL-LSM model and then the RL-THM model. Additionally, no significant changes in ROM were observed at any surgical segments under flexion across the three restorative laminoplasty models. When comparing the RL-HSM model with the RL-THM and RL-LSM models, it was observed that the largest ROM increase at all surgical segments occurred under axial rotation. There were no significant ROM variations between the RL-HSM and RL-LSM models under lateral bending, and no significant differences were found at L3/4 and L2/4 under extension. Additionally, the ROM variations between the RL-HSM and RL-THM models were not significant under extension and lateral bending at L3/4 but were significant at L2/3 and L2/4. When comparing the RL-LSM and RL-THM models, it was noted that the greatest increase in ROM at all surgical segments occurred under axial rotation. Conversely, there were no significant differences in ROM values between the two models under flexion, extension, or lateral bending. Comparing the displacements among the three restorative laminoplasty models, it was found that the maximum URES values under all directions in the RL-HSM model were smaller than those in the RL-THM model. Moreover, the maximum URES values under axial rotation and flexion were smaller than those in the RL-LSM model, while the maximum URES values under extension and lateral bending were similar to those in the RL-LSM model. Notably, the maximum URES values in the RL-LSM model were smaller than those in the RL-THM model, with exception of extension moments.

The conclusion reached in our study is that a restorative laminoplasty with the fixation of miniplates had a motion-limiting effect on a laminectomy in all motion planes, providing better stability in axial rotation and flexion/extension. Additionally, the HSMs demonstrated a superior performance compared to that of LSMs and THMs in maintaining the postoperative segmental stability of motion segments, particularly in axial rotation moments. We discovered that the restorative laminoplasty with the fixation of miniplates could enhance the stability, anti-rotation, and anti-flexion/extension capabilities of the spine. Upon analyzing the results, we concluded that the laminar miniplates had minimal impact on motion and displacement during lateral bending. This was primarily attributed to the fact that the motion plane of the spine during lateral bending was inconsistent with the direction of the restored musculatures and ligaments. A conventional laminectomy involves sacrificing the lamina, spinous process, and PLC, which can alter the biomechanical behavior of the spine. However, spinous processes can block each other, reducing the magnitude of extensions, while the PLC, which typically acts as a tension band during flexion, can resist flexion forces [46]. Additionally, the lamina can help resist axial rotation moments, and when combined with spinous processes and PLC, it may have a stabilizing effect on the spine in three dimensions [46]. Consequently, following a conventional laminectomy, the lumbar spine structure may become vulnerable to segmental instability at the operated segments, particularly in rotation and flexion/extension moments. As expected, in our study, we found that the laminectomy model exhibited greater levels of motion and displacement compared to those in the intact model at the impaired segments during axial rotation and flexion/extension moments. Theoretically speaking, preserving the central posterior osteo-ligamentous structure could provide a stabilizing effect for preventing post decompression instability and spondylolisthesis. Additionally, the miniplates system was implanted post laminectomy in a restorative laminoplasty to maintain the integrity of the spine by preserving the lamina, spinous process, and PLC. The continuity of the posterior structure in restorative laminoplasties is crucial for spinal integrity, anatomical restoration, functional recovery, and reducing instability after decompression surgery. Therefore, the motion, displacement levels, and ligament tensile forces in restorative laminoplasties were significantly enhanced due to varying degrees of posterior structure preservation. Higher levels of ROM and displacement after laminectomies indicate the preservation of a more dynamic structure of the spine. Factors that contribute to maintaining stability (such as bone nonunion and fibrosis unions) and those that weaken stability (such as the movement of the broken end of the lamina and the loss of the lamina, spinous process, and PLC) coexist post laminectomy, but the patient’s overall maintained capacity is often weak [54,78]. Restorative laminoplasties could hold the lamina in nearly its original position (before a solid bony fusion and a strong fibrous union are obtained). Therefore, the application of restorative laminoplasties with miniplate fixation reinforces maintenance factors and restores weakening factors, preserving and restoring the dynamic structures of the spine. As expected, in our study, the restorative laminoplasty with the fixation of miniplates demonstrated superiority over a conventional laminectomy by effectively restoring the biomechanical properties of posterior elements, allowing for the recovery of motion and limiting load displacement responses in the spine after surgery, particularly in axial rotation and flexion/extension moments.

Another conclusion drawn from this study was that HSMs in restorative laminoplasties are more effective than LSMs and THMs in preserving the motion and displacement of treated segments, especially in axial rotation moments. Upon analyzing our results, would like to expression the following opinions. In the field of orthopedic surgery, plates play a crucial role in bone fixation after traumatic injuries or osteotomies. These plates not only provide the rigid fixation and precise repositioning of fractured parts, but also apply compressive stress and strain at the fracture site to stimulate bone healing. During load bearing, the plates need to maintain the fractured ends in place while appropriately avoiding excessive motion and displacement to promote bone healing and bone density adaptation. The rationale behind plate instrumentation fixation is that the removal of posterior elements with a conventional laminectomy is destructive to the anatomy of spine and has been shown to alter its biomechanics. A material with suitable mechanical properties and biological activity as well as a design that morphologically matches complex lumbar structures are important for the function of laminar plates. Titanium alloy materials exhibit excellent levels of biocompatibility and mechanical strength, making them the preferred choice for bone implants. Furthermore, the design and properties of a bone plate must match the anatomical morphology and biomechanical requirements of the specific bone to achieve optimal fixation. Research has indicated that the slope angle of the lamina is 97.8 ± 3.0° at T9 and 129.0 ± 7.5° at L3 [79]. The slope angle of HSMs in our study exhibited arcs of 115° and 125°, with the length consistent with previous evidence. Normally, these HSMs could be directly applied to the majority of individuals and secured without the need for manual bending. However, in cases in which the laminar anatomy differs from the norm, a specialized miniature plate bender may be required to adjust the miniplates with minimal damage.

The variation in the results may be primarily attributed to the effects of spatial motion mechanics. THMs exhibited a nearly monohedral fixation, LSMs showed a nearly pyramidal fixation, and HSMs displayed a nearly prismatic fixation. Prismatic fixation is considered more effective than pyramidal fixation and monohedral fixation in avoiding rotation and displacement. This is attributed to its superior stress distribution in coplanar and coaxial configurations, thereby reducing instability caused by rotation and displacement. This study had certain limitations. It was a preliminary assessment of the biomechanical stability, efficacy, and feasibility of a restorative laminoplasty with the fixation of miniplates following a laminectomy. The focus of this study was on comparing the biomechanical effects of load motion and displacement across different restorative laminoplasty procedures, rather than comparing the fixation of miniplates with other techniques. To fully comprehend the value of the fixation of miniplates in restorative laminoplasties, it is imperative to assess the characteristics of miniplates from various angles. Our team has conducted a series of studies in recent years to explore the biomechanical implications of restorative laminoplasties. The next phase of our research will delve into the therapeutic mechanism, spatial motion mechanics, clinical efficacy, and prognosis of restorative laminoplasties. Additionally, further research is needed to investigate the impact of spinal degeneration, selection of fixed segments, and spinal defects post decompression on restorative laminoplasties. While this study was the first to use FE analysis models for restorative laminoplasties and points to subject-specific postsurgical changes in motion and displacement, additional FE analyses, animal or in vitro experiments, and clinical studies are necessary to evaluate the effects of restorative laminoplasties so as to provide further comprehensive evidence and guidance for their clinical application.

## 5. Conclusions

The evidence above suggests that the restorative laminoplasty with the fixation of miniplates had a motion-limiting effect on the laminectomy in all motion planes and yielded better levels of stability in axial rotation and flexion/extension. Additionally, the H-shaped miniplates were superior to the L-shaped and two-hole miniplates in maintaining the postoperative segmental stability of the motion segments, especially in axial rotation moments. Therefore, patients who are subjected to laminectomies should avoid excessive axial rotation and hyperflexion/extension, and patients undergoing restorative laminoplasties should be cautious of excessive axial rotation. The biomechanical findings obtained from this study provide quantitative data to support the development of more subtle surgical procedures for restorative laminoplasties in the future. Moreover, materials with appropriate mechanical properties and biological activity, along with designs morphologically matching complex lumbar structures, are crucial for the functionality of laminar miniplates. The aim of this study is to provide recommendations for the future design and clinical application of miniplates. The evidence indicates that using four-point stressed miniplates with two transverse holes positioned proximally and distally from the fracture site, which are specifically designed to align with the laminar anatomy, can lead to more effective laminoplasty procedures. Above all, the restoration of the overall posterior elements as much as possible, respecting the patient’s normal anatomy, should always be pursued.

## Figures and Tables

**Figure 1 bioengineering-11-00519-f001:**
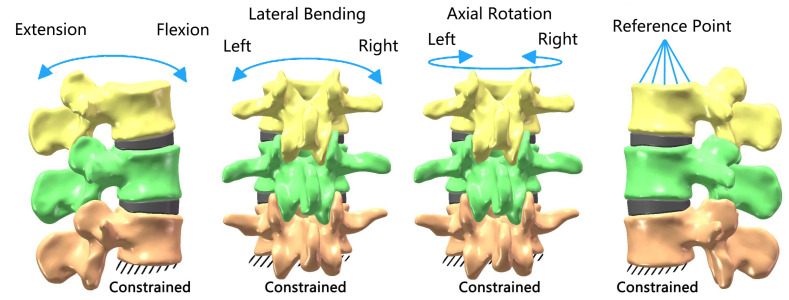
A schematic representation of the applied boundary and loading conditions on the L2/4 segments.

**Figure 2 bioengineering-11-00519-f002:**
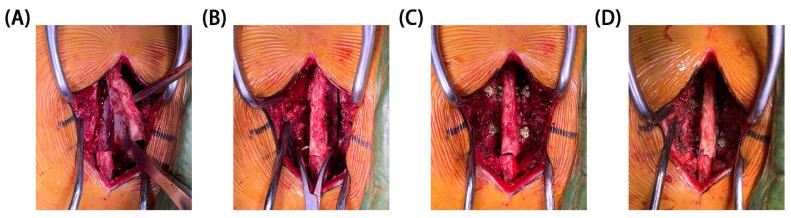
The surgical protocols of restorative laminoplasty with H-shaped miniplates (HSMs) fixation. Laminectomy with bilateral laminotomy and unilateral spinous ligament dissection (**A**), the posterior elements were placed in the desired position (**B**), fixation of HSMs in restorative laminoplasty after laminectomy (**C**), the repair of spinous ligaments after restorative laminoplasty (**D**).

**Figure 3 bioengineering-11-00519-f003:**
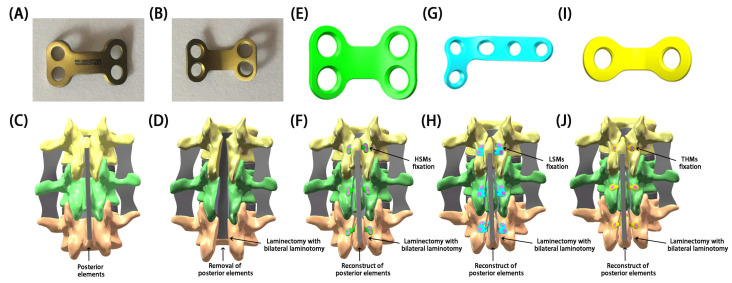
The physical pictures of HSM (**A**,**B**), the INT model (**C**), the LE model (**D**), the HSM model and the RL-HSM model (**E**,**F**), the LSM model and the RL-LSM model (**G**,**H**), and the THM model and the RL-THM model (**I**,**J**).

**Figure 4 bioengineering-11-00519-f004:**
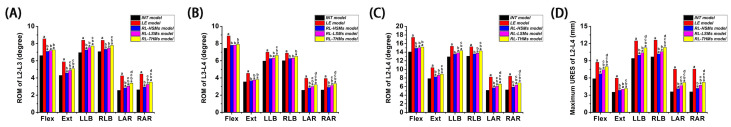
ROM and maximum URES in the five models under all motions. (**A**): The ROM of L2-L3. (**B**): The ROM of L3-L4. (**C**): The ROM of L2-L4. (**D**): The maximum URES of L2-L4. INT: intact. LE: laminectomy. RL-HSMs: restorative laminoplasty with H-shaped miniplates. RL-LSMs: restorative laminoplasty with L-shaped miniplates. RL-THMs: restorative laminoplasty with two-hole miniplates. Flex: flexion; Ext: extension; LLB: left lateral bending; RLB: right lateral bending; LAR: left axial rotation; RAR: right axial rotation; ROM: range of motion. a: A percentage increment (% increment) in ROM of more than 5% when compared with INT model. b: A percentage reduction (% reduction) in ROM of more than 5% when compared with LE model. c: A percentage change (% change) in ROM of more than 5% when compared with the RL-HSM model. d: A percentage change (% change) in ROM of more than 5% when compared with the RL-LSM model.

**Figure 5 bioengineering-11-00519-f005:**
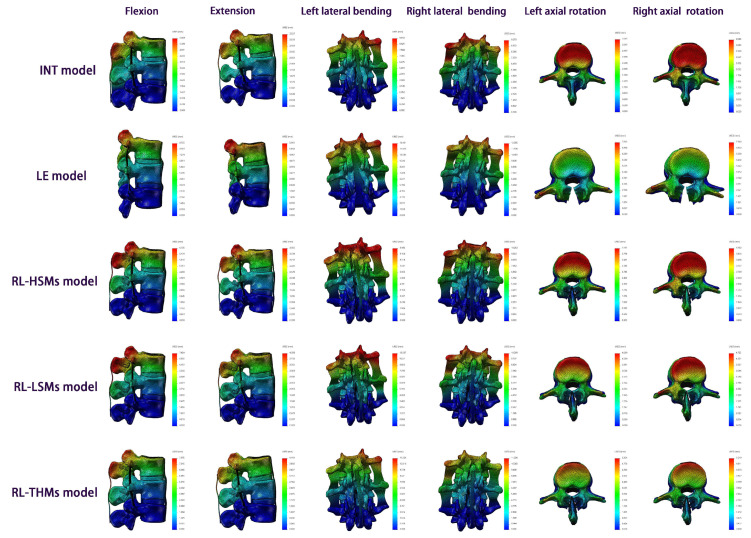
The URES (resultant displacement) distributions in INT model, LE model, RL-HSM model, RL-LSM model, and RL-THM model under all directions. In the visual representation, red represents large values of resultant displacement, while blue represents small values. The color bar on the right displays the corresponding resultant displacement value of the detail. The maximum URES corresponds to the highest resultant displacement value on the color bar.

**Table 1 bioengineering-11-00519-t001:** Properties of different components in the INT model.

Material	Young’s Modulus (MPa)	Poisson’s Ratio
Cortical bone [31]	12,000.00	0.30
Cancellous bone [31]	100.00	0.20
Posterior elements [31]	3500.00	0.25
Cartilage endplate [32]	25	0.25
Nucleus pulposus [31]	1.0	0.49
Annulus fibrosusr [31]		
Matrix	4.2	0.45
Outer fiber	550	0.30
Interlayer fiber	454	0.30
Inner fiber	357	0.30
Ligaments [33]		
ALL	20	0.30
PLL	70	0.30
LF	50	0.30
ISL	28	0.30
SSL	28	0.30
ITL	50	0.30
CL	20	0.30
Internal fixation [31]	11,000	0.31

ALL: anterior longitudinal ligament. PLL: posterior longitudinal ligament. ISL: interspinous ligament. SSL: supraspinous ligament. LF: ligamentum flavum. ITL: ligament intertransversarii. CL: capsular ligament.

**Table 2 bioengineering-11-00519-t002:** Elements and node numbers for different mesh resolutions.

Variable	Element Number	Node Number
INT model		
Mesh 3.0 mm	208,107	116,771
Mesh 2.5 mm	267,852	155,841
Mesh 2.0 mm	395,423	257,464
Mesh 1.0 mm	1,814,822	1,217,851
Miniplates and screws		
Mesh 1.5 mm	17,220	9002
Mesh 1.25	19,844	10,398
Mesh 1.0	28,860	15,567
Mesh 0.75 mm	47,966	26,967

**Table 3 bioengineering-11-00519-t003:** The number of elements and nodes used in this study.

Variable	Element Number	Node Number
INT model	395,423	257,464
LE model	333,860	219,381
RL-HSM model	424,977	274,588
RL-LSM model	441,821	286,870
RL-THM model	401,228	260,567

INT: intact. LE: laminectomy. RL-HSMs: restorative laminoplasty with H-shaped miniplates. RL-LSMs: restorative laminoplasty with L-shaped miniplates. RL-THMs: restorative laminoplasty with two-hole miniplates.

**Table 4 bioengineering-11-00519-t004:** Comparison of the present INT model with previous results of in vitro study (°).

Variable	Flexion	Extension	Left Lateral Bending	Right Lateral Bending	Left Axial Rotation	Right Axial Rotation
L2-3						
Yamamoto et al. [39]	6.5 (0.3)	4.3 (0.3)	7.0 (0.6)	7.0 (0.6)	2.2 (0.4)	3.0 (0.4)
INT model	6.60	4.30	6.95	7.06	2.56	2.63
L3-4						
Yamamoto et al. [39]	7.5 (0.8)	3.7 (0.3)	5.8 (0.5)	5.7 (0.3)	2.7 (0.4)	2.5 (0.4)
INT model	7.46	3.55	5.97	6.01	2.58	2.61

**Table 5 bioengineering-11-00519-t005:** The percentage increment (% increment) in ROM when compared to INT model (%).

Variable	Flexion	Extension	Left Lateral Bending	Right Lateral Bending	Left Axial Rotation	Right Axial Rotation
% increment in L2/3						
LE model	29.55	36.51	20.58	18.56	66.02	69.58
RL-HSM model	7.42	6.51	4.46	3.82	11.33	12.17
RL-LSM model	8.79	13.26	8.35	5.52	19.14	23.19
RL-THM model	10.15	18.14	11.08	10.20	32.46	33.84
% increment in L3-4						
LE model	19.03	27.89	17.09	14.48	53.49	50.96
RL-HSM model	4.56	3.66	4.86	4.33	10.47	11.49
RL-LSM model	4.83	5.92	5.86	4.83	16.28	19.92
LP-THM model	6.03	7.32	10.89	8.32	24.42	27.97
% increment in L2-4						
LE model	23.97	32.61	18.96	16.68	59.73	60.31
RL-HSM model	5.90	5.22	4.64	4.06	10.89	11.83
RL-LSM model	6.69	9.94	7.20	5.20	17.70	21.56
RL-THM model	7.97	13.25	10.99	9.33	28.42	30.92

**Table 6 bioengineering-11-00519-t006:** The percentage reduction (% reduction) in ROM when compared to LE model (%).

Variable	Flexion	Extension	Left Lateral Bending	Right Lateral Bending	Left Axial Rotation	Right Axial Rotation
% reduction in L2-3						
RL-HSM model	17.08	21.98	13.37	12.43	32.94	33.86
RL-LSM model	16.02	17.04	10.14	10.99	28.24	27.35
RL-THM model	14.97	13.46	7.88	7.05	20.21	21.08
% reduction in L3-4						
RL-HSM model	12.16	18.94	10.44	8.87	28.03	26.14
RL-LSM model	11.94	17.18	9.59	8.43	24.24	20.56
RL-THM model	10.92	16.08	5.29	5.38	18.94	15.23
% reduction in L2-4						
RL-HSM model	14.57	20.65	12.04	10.82	30.57	30.24
RL-LSM model	13.94	17.10	9.89	9.84	26.31	24.17
RL-THM model	12.91	14.60	6.70	6.30	19.60	18.33

**Table 7 bioengineering-11-00519-t007:** The percentage changes (% changes) in ROM comparing the RL-HSM model and the RL-LSM model (%).

Variable	Flexion	Extension	Left Lateral Bending	Right Lateral Bending	Left Axial Rotation	Right Axial Rotation
% changes _RL-HSMs_ in L2-3						
RL-LSM model	1.27	6.33	3.72	1.64	7.02	9.83
RL-THM model	2.54	10.92	6.34	6.14	18.98	19.32
% changes _RL-HSMs_ in L3-4						
RL-LSM model	0.26	2.17	0.96	0.48	5.26	7.56
RL-THM model	1.41	3.53	5.75	3.83	12.63	14.78
% changes _RL-HSMs_ in L2-4						
RL-LSM model	0.74	4.48	2.44	1.10	6.14	8.70
RL-THM model	1.95	7.63	6.07	5.07	15.81	17.06
% changes _RL-LSMs_ in L2-3						
RL-THM model	1.25	4.31	2.52	4.43	11.18	8.64
% changes _RL-LSMs_ in L3-4						
RL-THM model	1.15	1.33	4.75	3.33	7.00	6.71
% changes _RL-LSMs_ in L2-4						
RL-THM model	1.20	3.01	3.54	3.93	9.11	7.69

**Table 8 bioengineering-11-00519-t008:** The maximum URES values and their variations (mm or %).

Variable	Flexion	Extension	Left Lateral Bending	Right Lateral Bending	Left Axial Rotation	Right Axial Rotation
Maximum URES (mm)						
INT model	5.869	3.527	9.413	9.679	3.597	3.588
LE model	8.735	5.961	12.461	12.585	7.523	7.563
RL-HSM model	6.735	3.855	9.982	10.203	4.117	4.183
RL-LSM model	7.424	4.038	10.387	10.589	4.669	4.725
RL-THM model	7.975	4.149	11.254	11.326	5.204	5.249
% increment						
LE model	48.83	69.01	32.38	30.02	109.15	110.79
RL-HSM model	14.76	9.30	6.04	5.41	14.46	16.58
RL-LSM model	26.50	14.49	10.35	9.40	29.80	31.69
RL-THM model	35.88	17.64	19.56	17.02	44.68	46.29
% reduction						
RL-HSM model	22.90	35.33	19.89	18.93	45.27	44.69
RL-LSM model	15.01	32.26	16.64	15.86	37.94	37.52
RL-THM model	8.70	30.40	9.69	10.00	30.83	30.60
% changes _RL-HSMs_						
RL-LSM model	10.23	4.75	4.06	3.78	13.41	12.96
RL-THM model	18.41	7.63	12.74	11.01	26.40	25.48
% changes _RL-LSMs_						
RL-THM model	7.42	2.75	8.35	6.96	11.46	11.09

## Data Availability

The raw data supporting the conclusion of this article will be made available from the corresponding author.

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
