# Peer review of "Comparative Biomechanical Stability of the Fixation of Different Miniplates in Restorative Laminoplasty after Laminectomy: A Finite Element Study"

_bioengineering, 2024, doi:10.3390/bioengineering11050519_

Round 1
Reviewer 1 Report
Comments and Suggestions for Authors
Some minor corrections have been included into the text file as remarks.
The paper is dedicated to an interesting subject that is very important for surgical practice. The study is aimed an improvement of the restorative laminoplasty with fixation by miniplates after laminectomy by biomechanical comparisons of miniplates with different shape and dimensions for restorative laminoplasty based on the literature review and by numerical computations with finite element method. The FEM computations are typical for this type of problems but an application of a three-vertebrae FEM model with different types applied boundary and loading conditions and the miniplates of different shape and dimensions gives a new insight to the problem. The results of the study could help in decision making on the choice of the miniplane for the fixation in an individual patient and vertebral bone. The calculated results must be carefully checked on animal models for the validation purposes.
The conclusions are consistent with the evidence and arguments presented but the main conclusion on the motion-limiting effect of all the plates tested on laminectomy in all motion planes is too general and rather obvious. The suggestion for the patients who were subjected to laminectomy to avoid excessive axial rotation, hyperflexion and extension also is evident. The main question on the biomechanically-substantiated strategy to compute (with FEM) and elaborate of the patient-specific design of the miniplates remain unsolved. The Conclusion part is too short and general compare to the section Results. I strongly suggest to rework the Conclusions and made them more useful for medical doctors, biomedical engineers and researchers in biomechanics.
The references are adequate. The figures are of a good quality. The total text is too long and detailed; it much be more concentrated on the main issues. The data in the Tables are too extensive and difficult to understand in 3D representation; some reasonable shortage and image-based presentation would improve the work.
The choice of the Yamamoto’s model is not well substantiated, and its validation on the intact model is not convincing enough.
A set of minor misprints and comments are marked in the text.
I suggest acceptance after minor revision.

Author Response
Replies to Reviewer 1
The paper is dedicated to an interesting subject that is very important for surgical practice. The study is aimed an improvement of the restorative laminoplasty with fixation by miniplates after laminectomy by biomechanical comparisons of miniplates with different shape and dimensions for restorative laminoplasty based on the literature review and by numerical computations with finite element method. The FEM computations are typical for this type of problems but an application of a three-vertebrae FEM model with different types applied boundary and loading conditions and the miniplates of different shape and dimensions gives a new insight to the problem. The results of the study could help in decision making on the choice of the miniplane for the fixation in an individual patient and vertebral bone. The calculated results must be carefully checked on animal models for the validation purposes.
Answer:
Thanks very much for the reviewer's point and interpretation.
Given the non-reproducible and unpredictable nature of clinical operations and therapeutic effects, along with the need for further confirmation of the stability of restorative laminoplasty with miniplates fixation, it's necessary for us to conduct biomechanical properties on miniplates. It's also worth noting that the surgical technique and the design and efficacy of new implants should be evaluated through biomechanical experiments using in vivo (clinical research), in vitro (animal experiments and cadaver studies), and in silico modeling (finite element analysis) to corroborate such postoperative alterations in spine kinematics and kinetics.
In vitro, in vivo and in silico modeling biomechanical studies corroborate such postoperative alterations in the spine kinematics and kinetics. More intuitive data can be obtained from traditional cadaver research and animal experiments, which are also very important for biomechanical study. However, the biomechanical results in vivo and in vitro may vary due to differences in cadaveric specimens, animal samples, and patient characteristics. Meanwhile, it is important to note that cadaveric research, animal experiments, and clinical retrospective studies may yield uncertain results due to limitations such as non-repeatability, proof heterogenicity and large consumption. Image-based in vivo studies can quantify only the postoperative alterations in vertebral kinematics/motions. In vitro cadaver testing lacks the stabilizing effects of active muscles and thereby cannot truly reproduce in vivo behavior. In vitro cadaver testing also can’t simulate the postoperative changes in the body, especially for restorative laminoplasty. Because there are two distinct states whether the fusion of the lamina was achieved or not, and this difference may significantly affect the stability after surgery. Therefore, the reviewer's feedback on subsequent animal studies was of great clinical significance. Actually, in fact, we are in the process of planning an animal experiment, using a caprine model, to compare with laminectomy in the early (non-fusion state) and late stage (fusion state) respectively. Nonetheless, the issue of inter-specimen variability is always a significant concern when performing animal studies, presenting a challenge that must be addressed.
There has been limited progress in biomechanical research. There are many frequently-used kinds of restorative laminoplasty. The answer about relativity between choice of restorative laminoplasty techniques and result of stability is not given throughout. Based on this, FE analysis combined with model measurement research can be regarded as a reliable approach for evaluating the biomechanical characteristics of different miniplates techniques. The FE analysis is a valuable tool in the field of biomechanics and has been used to study spine biomechanics in the last decades. FE analysis is one of the best methods to reduce subject-related variability and create an idealized spine model, enabling realistic simulation of surgical procedures and the biomechanical impact of prostheses on the body. This method has been viewed as a reliable approach for evaluating the biomechanical stability of different internal fixation system, while also enabling evaluation of the biomechanical consequences of various spinal surgeries through the calculation and analysis of parameters such as range of motion (ROM) and displacement, among other. More and more scholars have begun to apply FE analysis to explore mechanical alterations in the physiological and pathological processes of the spine, as well as the operational principle, immediate postoperative stability, and stress distribution characteristics of various internal fixation devices in order to provide a basis for clinical improvement and optimization of the surgical plan of the spine.
Due to the complexity in vivo, we simplified the model in the process. Therefore, the FE results should be considered to have a similar tendency to the actual situation and provide a possible consequence in clinical settings but not present the same mechanical behavior as in vivo. Our results of the immediate postoperative stability of laminectomy in the lumbar spine aligns with previous in vitro cadaver findings and finite element studies. However, there is a lack of biomechanical comparisons in the literature regarding different miniplates used in restorative laminoplasty. To address this gap, our research aims to investigate the biomechanical effects of restorative laminoplasty with miniplate fixation following laminectomy in the lumbar spine, making it the first study of its kind.
It is difficult to perform reproducible experimental investigations or to apply physiological loads when using cadaver specimens. The FE method allows the calculation of stresses, strains and movements in the different structures involved. The advantage of the analytical over the experimental approach is that no new specimens are needed to modify particular parameters such as the degree of resection, the loads or the boundary conditions. The process of comparing numerical to experimental data and subsequently adjusting the computer model makes the finite element method a powerful tool for analyzing such biomechanical problems, as other studies have shown. It was a preliminary assessment of the biomechanical stability, efficacy, and feasibility of restorative laminoplasty with miniplates fixation following laminectomy. The focus of our study was on comparing the biomechanical effects of load motion and displacement across different restorative laminoplasty procedures. To fully understand the value of miniplates fixation in restorative laminoplasty, it is crucial to thoroughly evaluate the characteristics of the miniplates from different perspectives. The present study concentrated on finite element analysis exclusively and did not incorporate a cadaver or animal study. It is essential to verify the computed results on cadaver or animal models for validation purposes. These avenues will be explored in our future research endeavors, and we have already initiated these studies, with the experiment currently in the preparation phase.
One disadvantage to the current finite element model is that it does not account for variation in soft tissues forces at any segment that might arise because of surgical manipulation. Soft tissues such as muscles, fascia, and fat were not included in the simulation, with only the lumbar vertebrae, ligaments, and intervertebral discs being modeled. While these soft tissues provide slight traction due to their elasticity, there is currently no evidence to support the notion that this traction force affects biomechanical stability after lumbar surgery[1]. What’s more, it is commonly acknowledged that early stages of lumbar spine surgery often necessitate restricting lumbar muscle activity. Consequently, even in the absence of simulating soft tissues like muscles, the outcomes are unlikely to be affected[1].
The conclusions are consistent with the evidence and arguments presented but the main conclusion on the motion-limiting effect of all the plates tested on laminectomy in all motion planes is too general and rather obvious. The suggestion for the patients who were subjected to laminectomy to avoid excessive axial rotation, hyperflexion and extension also is evident. The main question on the biomechanically-substantiated strategy to compute (with FEM) and elaborate of the patient-specific design of the miniplates remain unsolved. The Conclusion part is too short and general compare to the section Results. I strongly suggest to rework the Conclusions and made them more useful for medical doctors, biomedical engineers and researchers in biomechanics.
Answer:
Thanks very much for the reviewer's point and interpretation.
Orthopedic surgeons have recently used L-shaped miniplates (LSMs) derived from metacarpal and phalanx or metatarsal miniplates for restorative laminoplasty in the spine, whereas neurosurgeons have typically employed cranial two-hole miniplates (THMs) for restorative laminoplasty procedures. While these miniplates are frequently utilized in clinical settings, the extensive shaping and bending needed during surgery may compromise their physiological and mechanical properties. This could result in prolonged surgical time, disruption of the surgical process, and potentially unnecessary effects. Furthermore, there is a risk of excessive fatigue or even fracture at the molding area, which could lead to severe complications.
During the initial phases of restorative laminoplasty at our institution, we utilized LSMs and T-shaped miniplates (TSMs) derived from metacarpal and phalanx or metatarsal miniplates, and cranial two-hole or four-hole miniplates (THMs or FHMs). The cranial THMs and FHMs are all miniplates with a linear shape. Based on our experience, it is advisable to use four-hole plates (LSMs and TSMs) for optimal fixation. These miniplates should secure the top and bottom lamina, while the middle lamina can be fixed with THMs alone. This approach ensures a stable and symmetrical fixation of titanium nails on both sides. However, we observed that when using four-hole plate miniplates (FHMs, TSMs, and LSMs), the longitudinal two-hole screws near the spinous end were prone to collision and loosening due to the limited laminar space. Likewise, the longitudinal two-hole screws placed at the laminar end could not be securely fastened due to the limited laminar length. To enhance the rotational stability and fixation effect of THMs, FHMs, TSMs, and LSMs, and minimize the biomechanical damage from bending and shaping of these plates, we ultimately created H-shaped miniplates (HSMs) for clinical application in reconstructing the posterior elements. These HSMs, tailored for restorative laminoplasty, were specifically designed based on the physiological characteristics and anatomical structure of the lamina. Clinical experience at our institution with HSMs in restorative laminoplasty has demonstrated their efficacy in treating intraspinal lesions.
A material with appropriate mechanical properties and biological activity, along with a design morphologically matching with the complex lumbar structures, is crucial for the functionality of laminar miniplates. The aim of this research is not only to demonstrate the superiority of our miniplates over the existing clinical miniplates in preserving spinal stability, but also to offer suggestions for future miniplates design and clinical implementation. Based on the results, we propose the use of four-point stressed miniplates with two transverse holes positioned proximally and distally from the fracture site, specifically designed to match the laminar anatomy, for more effective laminoplasty procedures.
The biomechanical stability of miniplates fixation can be elucidated by considering different types of traffic vehicles. Linear two-wheeled bicycles and triangular three-wheeled vehicles are less stable and more prone to rollover during U-turns, while quadrilateral four-wheeled vehicles are relatively stable in U-turns. However, even quadrilateral four-wheeled vehicles may still pose a risk of rollover during fast U-turns. When it comes to 90-degree turns, there is not a significant difference between the stability of two-wheeled, three-wheeled, and four-wheeled vehicles. In terms of forward braking and reversing, four-wheeled vehicles exhibit better balance and stability compared to two-wheeled bicycles and three-wheeled vehicles. The comparison of vehicle maneuvers to human body movements provides a unique perspective, where U-turns can be likened to axial rotation, 90-degree turns to lateral bending, and braking/reversing to flexion/extension. It is widely recognized that traffic vehicles exhibit better stability when moving forward compared to when reversing. This observation aligns with the findings of this study, which indicating that miniplates fixed in restorative laminoplasty procedures under flexion demonstrate better stability than those under extension.
The conclusions and manuscript have been revised as requested.
The references are adequate. The figures are of a good quality. The total text is too long and detailed; it much be more concentrated on the main issues. The data in the Tables are too extensive and difficult to understand in 3D representation; some reasonable shortage and image-based presentation would improve the work.
Answer:
Thanks very much for the reviewer's point and interpretation. The manuscript and the image-based presentation have been revised as requested. The table containing quantitative values of ROM and maximum URES was removed and replaced with images. However, the table displaying percentage variation was kept. This allows readers to easily access specific percentage variation values when needed.
The choice of the Yamamoto’s model is not well substantiated, and its validation on the intact model is not convincing enough.
Answer:
Thanks very much for the reviewer's point and interpretation. The Yamamoto’s model is one of the most widely used validation models in finite element analysis, which is referred to in many literatures[2-13]. The presented validation was performed on the basis of the experimental data selected by the authors. However, more experimental data are available in the literature (Guan et al., 2007; Niosi et al., 2008; Pearcy, 1985; Yamamoto et al., 1989) that can be used in validation[14]. The authors selected experimental studies conducted for a male subject due to the tested male model of the lumbar spine.
The ROM of current intact L2/4 model under six directions was tested and validated, using results from Yamamoto et al.’s in vitro cadaveric tests for the same loading conditions (Table 2). At L2/3, the differences between the intact FE model and the literature data were found to be within 3.08 % in flexion, 2.79 % in extension, 0.71 % in left lateral bending, 0.86 % in right lateral bending, 3.18 % in left axial rotation, and 3.67 % in right axial rotation; Similarly, at L3/4, the differences were within 3.20 % in flexion, 2.97 % in extension, 2.07 % in left lateral bending, 2.81 % in right lateral bending, 0.74 % in left axial rotation, and 0.80 % in right axial rotation. Our results indicating that the current intact L2/4 FE model was successfully constructed and could be used for further biomechanical analysis. Since all the data were conformed through normal human body parameters in vitro cadaveric tests, the intact FE model was able to replicate human physiological movement of L2/4 vertebrae.
A set of minor misprints and comments are marked in the text.
Answer:
Thanks very much for the reviewer's point and interpretation. Due to personal commitments related to my applications for the National Natural Science Foundation of China (NSFC) and the National Health Technology Promotion Project on the restorative laminoplasty, there was a delay in writing and completing the initial draft. Regrettably, in a rush, the manuscript was submitted without a thorough checking before the deadline, resulting in some minor issues that need to be rectified.
The minor misprints and comments in manuscript have been revised as requested.
The comments in the text “Does it mean, neither ligaments nor muscles were introduced in the FEM model? What type of the connection between the vertebrae were used for the FEM computations?”
Answer: To better highlight the image, the ligament was not included in this particular figures. However, in other displayed figures, the ligaments are shown. The muscles were not introduced in the FEM model. Soft tissues such as muscles, fascia, and fat were not included in the simulation, with only the lumbar vertebrae, ligaments, and intervertebral discs being modeled. While these soft tissues provide slight traction due to their elasticity, there is currently no evidence to support the notion that this traction force affects biomechanical stability after lumbar surgery[1]. What’s more, it is commonly acknowledged that early stages of lumbar spine surgery often necessitate restricting lumbar muscle activity. Consequently, even in the absence of simulating soft tissues like muscles, the outcomes are unlikely to be affected[1].
The comments in the text “1 person is not enough for a ststiatically relevant confirmation”
Answer: The percentage variation in different procedures was compared, and a difference of more than 5% was considered significant[15].
I suggest acceptance after minor revision.

Reviewer 2 Report
Comments and Suggestions for Authors
Involved is the combined expertise of 10 authors associated with 3 orthopaedic departments, a research testing centre and an outpatients department in China.
The gross anatomy of the spine is clinically complex and presents a biomechanical challenge for orthopaedic surgeons. The prime aim of the investigation is the restoration of normal form and function, as much as is possible, after invasive surgery via the posterior aspect of the vertebral column. As a model system a single, healthy, 30 yr old male volunteer is CT scanned to provide a finite element 3D model of the normal lumbar spine consisting of 3 lumbar vertebrae, 2 intervertebral discs and 7 associated ligaments. This was used to test restorative laminoplasty and the efficacy of laminectomy fixation comparing three designs of miniplate of which the H-shaped version proved superior in mitigating stability.
Background to the investigation is a substantial detailed history of the twin surgical interventions of laminectomy and laminoplasty and their respective advantages and disadvantages, together with the benefit of structurally stabilising minipins of varying design.
Considerable commitment is indicated in addressing subject-specific postsurgical changes in motion and displacement, with an ambition for future FE analysis via animal or in vitro experiments for more comprehensive guidance in clinical application.
1.Title: Appropriate. Miniplates ……singular not plural, i.e., with miniplate fixation?
2.For the benefit of readers who are not gross anatomists a figure at the outset clearly identifying the vertebral body “lamina” would avoid uncertainty, as for histologists the term applies to other structures e.g., the basement membrane.
3. In selecting a youthful male model, are there any thoughts about sex-related factors such as those differentiating cancellous bone substructure in ageing men and postmenopausal women with the apparent prospect of strategic “crumple zones” in the latter?
4. The paragraphs are heavy on detail but logically assembled and well written. It would benefit from minor editorial attention throughout, examples of which are as follows:
Lines 37 and 41. “Our study suggests”, also “ in our study”. It is generally more authoritative to be impersonal throughout, e.g., “the evidence suggests”.
Line 111, 134, 143, 149. Avoid the frequent apostrophe which is misplaced in formal scientific writing: they’re and there’s should be “they are” and “there is”, and it’s should be “ it is”.
Line 122. Uptill ; is not English and should be until.
Line 582. …… preservation. Their………
Line 788. Furthermore, moreover. Misuse of two words where one will do.
Line 959. We believe. No, no, no………….The evidence shows……
Line 985. “The findings of our study” “The evidence above suggests” is better.
Comments on the Quality of English LanguageGenerally very good as indicated in comments to authors.
Author Response
Replies to Reviewer 2
Involved is the combined expertise of 10 authors associated with 3 orthopaedic departments, a research testing centre and an outpatients department in China.
. The prime aim of the investigation is the restoration of normal form and function, as much as is possible, after invasive surgery via the posterior aspect of the vertebral column. As a model system a single, healthy, 30 yr old male volunteer is CT scanned to provide a finite element 3D model of the normal lumbar spine consisting of 3 lumbar vertebrae, 2 intervertebral discs and 7 associated ligaments. This was used to test restorative laminoplasty and the efficacy of laminectomy fixation comparing three designs of miniplate of which the H-shaped version proved superior in mitigating stability.
Background to the investigation is a substantial detailed history of the twin surgical interventions of laminectomy and laminoplasty and their respective advantages and disadvantages, together with the benefit of structurally stabilising minipins of varying design.
Considerable commitment is indicated in addressing subject-specific postsurgical changes in motion and displacement, with an ambition for future FE analysis via animal or in vitro experiments for more comprehensive guidance in clinical application.
1.Title: Appropriate. Miniplates ……singular not plural, i.e., with miniplate fixation?
Answer:
Conventional midline laminectomy with bilateral laminotomy and removal of the posterior elements is a standard procedure for the removal of herniated disks and decompression of the spinal canal and entrapped nerve roots. Since bilateral laminotomy is performed during the laminectomy, at least two plates are needed for reconstructing the posterior elements. Thus, we suggest using the term 'miniplates' in the plural form.
2.For the benefit of readers who are not gross anatomists a figure at the outset clearly identifying the vertebral body “lamina” would avoid uncertainty, as for histologists the term applies to other structures e.g., the basement membrane.
Answer:
Thanks very much for the reviewer's point and interpretation.
The “lamina” is an anatomical term which is commonly used in the fields of neurosurgery and orthopedics. We annotated the lamina where it first appeared, specifically referring to the lamina as the vertebral plate. Furthermore, it is worth mentioning that a laminectomy is actually a lamina-ectomy, and a laminotomy is actually a lamina-otomy.
- In selecting a youthful male model, are there any thoughts about sex-related factors such as those differentiating cancellous bone substructure in ageing men and postmenopausal women with the apparent prospect of strategic “crumple zones” in the latter?
Answer:
Thanks for the reviewer's point and interpretation. A numerical model of the lumbar spine was developed based on the data available in the literature. To ensure the reliability of the model, validation based on experimental data selected by the authors is necessary. More experimental data are available in the literature that can be used in validation. However, the authors all selected experimental studies conducted for a male subject due to the tested male model of the lumbar spine[14]. Therefore, adult males were chosen for the study. However, given the population's biodiversity, it would be necessary to verify and validate the obtained data with other experimental studies. Considering the wide range of geometric and material characteristics in the human body and still a limited amount of validation data, it is essential to fully share the results that can be obtained with the FEM. Future research should also consider sex-related factors, such as those differentiating cancellous bone substructure in ageing men and postmenopausal women with the apparent prospect of strategic “crumple zones” in the latter?
- The paragraphs are heavy on detail but logically assembled and well written. It would benefit from minor editorial attention throughout, examples of which are as follows:
Lines 37 and 41. “Our study suggests”, also “ in our study”. It is generally more authoritative to be impersonal throughout, e.g., “the evidence suggests”.
Answer:
Thanks for the reviewer's point and interpretation. The issues in manuscript have been revised as requested.
Line 111, 134, 143, 149. Avoid the frequent apostrophe which is misplaced in formal scientific writing: they’re and there’s should be “they are” and “there is”, and it’s should be “ it is”.
Answer:
Thanks for the reviewer's point and interpretation. The issues in the manuscript have been revised as requested.
Line 122. Uptill ; is not English and should be until.
Answer:
Thanks for the reviewer's point and interpretation. The 'Up till now' has been revised as 'To date'.
Line 582. …… preservation. Their………
Answer:
Thanks for the reviewer's point and interpretation. The issues in the manuscript have been revised as requested.
Line 788. Furthermore, moreover. Misuse of two words where one will do.
Answer:
Thanks for the reviewer's point and interpretation. The issues in the manuscript have been revised as requested.
Line 959. We believe. No, no, no………….The evidence shows……
Answer: Thanks for the reviewer's point and interpretation. The issues in the manuscript have been revised as requested.
'We believe that the discrepancy in results can be mainly attributed to the effects of spatial motion mechanics' has been revised as 'The variation in the results may be primarily attributed to the effects of spatial motion mechanics'.
Line 985. “The findings of our study” “The evidence above suggests” is better..
Answer:
Thanks for the reviewer's point and interpretation. The issues in the manuscript have been revised as requested.

Reviewer 3 Report
Comments and Suggestions for Authors
The manuscript evaluates the biomechanical effects of different miniplate shapes used in restorative laminoplasty after laminectomy. The study uses finite element analysis to compare stability, motion limitation, and displacement between designs using H-shaped, L-shaped, and T-shaped miniplates.
The presentation of the work is methodical and detailed, particularly in describing the finite element modeling process and the biomechanical evaluations performed.
Although finite element analysis in spinal surgery is nothing new, the specific comparative evaluation of different miniplate designs in the context of post-laminectomy restorative laminoplasty provides new insights, particularly regarding biomechanical performance differences between plate designs. miniplates.
The focus on comparing three different types of miniplates for restorative laminoplasty in a single study adds insight to existing research, which instead often focuses on broader comparisons of surgical techniques without detailed analysis of hardware variations.
The results obtained highlight the peculiarities of the individual plates, this could guide the surgical decision-making process on the choice of miniplate shapes.
FE models are built on detailed and precise data, ensuring that simulations are relevant and targeted to the study objectives. The accuracy in the construction and use of these models confirms the reliability of the conclusions.
The manuscript is well written in clear English.
The bibliography is updated and relevant, there are fundamental and recent references that underline the contextual position of the manuscript within the ongoing research.
The images and tables are of high quality and provide clear and informative visual support effectively illustrating the differences in biomechanical performance between the models studied.
Author Response
Replies to Reviewer 3
The manuscript evaluates the biomechanical effects of different miniplate shapes used in restorative laminoplasty after laminectomy. The study uses finite element analysis to compare stability, motion limitation, and displacement between designs using H-shaped, L-shaped, and T-shaped miniplates.
The presentation of the work is methodical and detailed, particularly in describing the finite element modeling process and the biomechanical evaluations performed.
Although finite element analysis in spinal surgery is nothing new, the specific comparative evaluation of different miniplate designs in the context of post-laminectomy restorative laminoplasty provides new insights, particularly regarding biomechanical performance differences between plate designs. miniplates.
The focus on comparing three different types of miniplates for restorative laminoplasty in a single study adds insight to existing research, which instead often focuses on broader comparisons of surgical techniques without detailed analysis of hardware variations.
The results obtained highlight the peculiarities of the individual plates, this could guide the surgical decision-making process on the choice of miniplate shapes.
FE models are built on detailed and precise data, ensuring that simulations are relevant and targeted to the study objectives. The accuracy in the construction and use of these models confirms the reliability of the conclusions.
The manuscript is well written in clear English.
The bibliography is updated and relevant, there are fundamental and recent references that underline the contextual position of the manuscript within the ongoing research.
The images and tables are of high quality and provide clear and informative visual support effectively illustrating the differences in biomechanical performance between the models studied.
Answer:
The reviewer' recognition of this research is greatly appreciated. A material with appropriate mechanical properties and biological activity, along with a design morphologically matching with the complex lumbar structures, is crucial for the functionality of laminar miniplates. The objective of this study is to provide recommendations for the future design and clinical application of miniplates. Based on the results, we propose the use of four-point stressed miniplates with two transverse holes positioned proximally and distally from the fracture site, specifically designed to match the laminar anatomy, for more effective laminoplasty procedures.
For limb fracture, yield strength and stress distribution improved when a triangular or alternating pattern of screws was used. There was no effect on axial stiffness when more than three screws were used proximally and distally from the fracture. Torsional rigidity did not increase with more than four screws on both sides of the fracture[16]. Following surgery for limb fractures, it is commonly recommended to restrict weight-bearing and rotation, sometimes resorting to immobilization with a plaster brace. Patients with fractures generally comply well with post-surgery weight-bearing restrictions. Furthermore, engaging in normal activities without placing weight on the limbs has little impact on the mobility of the fractured limb ends. Given the limited rotational activity of limb fractures, the primary factor to consider when designing internal fixation for limb fractures is the mechanical characteristics of the plate.
Spinal surgery patients experience different mechanical effects compared to conventional fractures. The complex biomechanical changes were also associated with complex spinal movements such as flexion, extension, bending, and rotation, because of that different mechanical load distribution occured in different postures. Despite efforts to immobilize the spine with a brace, spinal movements inevitably occur during daily activities due to the spine's complexity. Prolonged immobilization post-surgery can result in lumbar muscle atrophy, making it advisable to engage in appropriate lumbar back muscle functional exercises during bed rest. However, these exercises may heighten the risk of jiggle of fracture end in the posterior column and lamina due to increased mechanical load at those sites. Rotational activities tend to have a significant impact on the spine, and such movements are often unavoidable in daily tasks. Therefore, when considering internal fixation for spine, it is crucial to consider not only the mechanical properties of the steel plate but also its flexibility. This explains why plates used for spine are thinner and more flexible than those used for limb fractures.
The use of two-hole plates, like THMs in our study, with two points of stress have a limited anti-rotation effect and are prone to fracture retraction, collapse, and door-closing effect. This limitation becomes more pronounced in wider fractures, such as those at the lamina site. On the other hand, the triangular configuration fixed plates, such as LSMs in our study, are designed based on the principle of triangular stress points, offering excellent stabilization for axial movement, sagittal movement, forward and backward movement and lateral bending movement. However, during rotational activities, its stability is compromised due to the axial single surface and strip fixation, making it prone to deformation. In contrast, the four-point stressed plates like HSMs in our study, are fixed following a four-point spatial stress principle, enabling the formation of multiple triangles and multi-facet fixation in the axial direction when rotating, providing better stability during rotation.
The rationale behind plate fixation in laminar fractures can be elucidated by examining the management of rotationally unstable fractures, such as scapula and pelvic fractures. Therefore, in the internal fixation procedure for these unstable fractures, multiple plates are usually employed for stabilization in close proximity to the fracture site, with more than three screws (a minimum of four screws) used proximally and distally from the fracture.

Reviewer 4 Report
Comments and Suggestions for Authors
Undoubtedly, this is a sound, timely paper that presents important contributions on the effects of different miniplates use in a restorative laminoplasty using a numerical (FEA) approach. The research was carefully and adequately framed and it is presented in a transparent and comprehensive way. Additionally, a solid and robust discussion of results is presented. These are the strengths of the research. The manuscript is also well addressed and structured. There are, however, several weaknesses in the manuscript, which I consider important. A non-exhaustive list follows:
1) Since a FE numerical model has been built, why images that represent the CAD model developed (3D solid geometric model) aren' t shown? It will be interesting to show an image containing the constraints (boundary) and loaded conditions considered.| Please revise;
2) The data in Table 1, what bibliographic source do they come from? | Please revise;
3) In contact simulations, the friction coefficients considered where chosen based in what reference? | Please revise;
4) The axial preload and torque values considered where chosen based in what reference? | Please revise;
5) Since a mesh sensitivity and convergence test were done, why a graphic or table aren’t shown to confirm the results described? Please revise;
6) Figures 3c), 3d), 3f), 3h and 3j) needed to be highlighted | Please revise;
7) Equations presentation (page 10) must be improved | Please revise;
8) The paper will much benefit if a comparison between main results and results achieved by other research works is conducted, for example in section “Validation of the Intact Model” the reference Yamamoto, et al. is overly used | Please revise;
9) Why the URES differences between LE model and intact model were so high? | Please revise;
10) Why stress distribution fields weren't shown? | Please revise;
11) Figure 4 presentation needs to be rethink| Please revise;
12) The results discussion is an assembly between a revision work and the results presentation obtained here by FEA | Please revise;
Suggestion: rethink the paper title, a more “clean” will be welcome!!
Author Response
Replies to Reviewer 4
Undoubtedly, this is a sound, timely paper that presents important contributions on the effects of different miniplates use in a restorative laminoplasty using a numerical (FEA) approach. The research was carefully and adequately framed and it is presented in a transparent and comprehensive way. Additionally, a solid and robust discussion of results is presented. These are the strengths of the research. The manuscript is also well addressed and structured. There are, however, several weaknesses in the manuscript, which I consider important. A non-exhaustive list follows:
1) Since a FE numerical model has been built, why images that represent the CAD model developed (3D solid geometric model) aren' t shown? It will be interesting to show an image containing the constraints (boundary) and loaded conditions considered.| Please revise;
Answer:
Thanks for the reviewer's point and interpretation.
CAD is a technology commonly utilized in drafting and design. It enables designers to create precise, intricate, and modifiable graphics using computers. However, CAD files cannot be directly shared for display. To address this, CAD drawings must be converted into image formats for accessibility on various platforms and for sharing with others. By converting CAD files to high-quality image formats, they can be easily opened, edited, and shared using common image viewers or editors. This enhances the usability of the images across different devices and platforms.
2) The data in Table 1, what bibliographic source do they come from? | Please revise;
Answer:
Thanks for the reviewer's point and interpretation. The data in Table 1 was come from previous published literature, the issues in Table 1 have been revised as requested.
Table 1. Properties of different components in the model.
Material |
Young’s modulus (MPa) |
Poisson's ratio |
Cortical bone[17] |
12000.00 |
0.30 |
Cancellous bone[17] |
100.00 |
0.20 |
Posterior elements[17] |
3500.00 |
0.25 |
Cartilage endplate[11] |
25 |
0.25 |
Nucleus pulposus[17] |
1.0 |
0.49 |
Annulus fibrosusr[17] |
|
|
Matrix |
4.2 |
0.45 |
Outer fiber |
550 |
0.30 |
Interlayer fiber |
454 |
0.30 |
Inner fiber |
357 |
0.30 |
Ligaments[18] |
|
|
ALL |
20 |
0.30 |
PLL |
70 |
0.30 |
LF |
50 |
0.30 |
ISL |
28 |
0.30 |
SSL |
28 |
0.30 |
ITL |
50 |
0.30 |
CL |
20 |
0.30 |
internal fixation[17] |
11000 |
0.31 |
ALL: anterior longitudinal ligament. PLL: posterior longitudinal ligament. ISL: interspinous ligament. SSL: supraspinous ligament. LF: ligamentum flavum. ITL: ligament intertransversarii. CL: capsular ligament.
3) In contact simulations, the friction coefficients considered where chosen based in what reference? | Please revise;
Answer:
The issues in the manuscript have been revised as requested.
The boundary and loading conditions were set according to previous studies[7, 10, 12, 19, 20]. A 3D surface-to-surface sliding contact with friction was specified to simulate the joints contact behavior with a finite sliding interaction defined to allow random motions, including sliding, rotation, and separation. Articulating friction was neglected and only transmitted normal forces were considered. In the interaction settings, cortical bone was bound to cancellous bone, ligaments were bound to the outer surface of cortical bone. The contact between facet joints was defined as nonlinear, surface-to-surface, frictionless sliding contact elements[10, 20]. The contact between vertebra and IVDs was designated as mutual contact with a friction coefficient of 0.08[7]. The contact type of miniplate and lamina was set as friction with a friction coefficient of 0.2[5]. The contact type of screw and lamina was set to binding mode[5, 12].
4) The axial preload and torque values considered where chosen based in what reference? | Please revise;
Answer:
The boundary and loading conditions used during the analysis were derived from Yamamoto’s in vitro studies[21]. The maximum loading condition of 10 Nm is a widely accepted parameter in finite element analysis and cadaver research. The maximum loading protocol, including axial preload and torque values, was designed to simulate maximum physiologic motions without causing harm to the segments. Hence, we opted for the maximal loading conditions of 10 Nm in our study.
5) Since a mesh sensitivity and convergence test were done, why a graphic or table aren’t shown to confirm the results described? Please revise;
Answer:
The issues in mesh sensitivity and convergence test have been revised as requested.
6) Figures 3c), 3d), 3f), 3h and 3j) needed to be highlighted | Please revise;
Answer:
The issues in Figures 3c), 3d), 3f), 3h and 3j have been highlighted.
7) Equations presentation (page 10) must be improved | Please revise;
Answer:
The issues in Equations presentation have been revised as requested.
8) The paper will much benefit if a comparison between main results and results achieved by other research works is conducted, for example in section “Validation of the Intact Model” the reference Yamamoto, et al. is overly used | Please revise;
Thanks very much for the reviewer's point and interpretation.
To validate the intact FE model, kinematics data from the present intact FE model were compared with in vitro experimental data obtained from cadavers as reported by Yamamoto et al[21]. the reference Yamamoto, et al.[21] is overly used in FE analysis[2-13]. The results of the laminectomy model were compared with findings from other research works in the manuscript. This study stands out as the initial attempt to investigate the biomechanical stability of different miniplates fixation in restorative laminoplasty, shedding light on subject-specific postsurgical alterations in motion and displacement. As a result, further comparison of the main outcomes of restorative laminoplasty was not carried out.
9) Why the URES differences between LE model and intact model were so high? | Please revise;
Answer:
Thanks very much for the reviewer's point and interpretation.
In our study, compared to intact state, the laminectomy state showed the highest increase in ROM and displacement. Following laminectomy, the ROM increased at surgical segment (L2/3, L3/4 and L2/4) by 67.82 %, 52.22 %, and 60.02 % in axial rotation, 36.51 %, 27.89 %, and 32.61 % in extension, 29.55 %, 19.03 %, and 23.97 % in flexion, 19.56 %, 15.78%, and 17.81 % in lateral bending respectively. These findings align with previous in vitro results. Quint et al[22] discovered a substantial increase in ROM during laminectomy under submaximal loading conditions of 7.5 Nm. They observed a 117.4% increase in axial rotation, 35.0 % in extension, 32.0 % in flexion, 14.3 % in lateral bending. Lang et al.[22] tested FHCLS specimens under maximal loading conditions and reported even greater increases in ROM. With a 206.25% increase in axial rotation, 60.98 % in extension, 16.42 % in flexion, and 30.77% in lateral bending. Schmoelz et al.[23] conducted a study on FHCLS specimens (7.5 Nm) post laminectomy, revealing that laminectomy significantly increased the ROM in both treated segments (L4/5 and L3/4) to 187% and 234% of the intact specimen under axial rotation. Furthermore, under flexion/extension, laminectomy led to a 135% increase in ROM in both segments, while under lateral bending, it resulted in approximately 125% of the intact state. In the in vitro study (10 Nm) conducted by Strauss et al.[24], the ROM at L4/5 were increased by 31.12 % under axial rotation, 23.64 % under flexion/extension, and 2.62 % under lateral bending. The above studied concluded that laminectomy displaced the balance point ventrally. The instability observed post-laminectomy in our present study was also consistent with findings reported by Fuchs et al.[25], Hamasaki et al.[26], Grunert et al.[27], and Smith et al [28] in their in vitro studies, and by Zander et al. [29], Bresnahan et al.[30] and Liu et al.[31] in their FE studies.
Biplanar stereometry method is an accepted practice and was used to duplicate the in vitro (Figure 2). In this paper, the conventional biplanar stereoscopic measurement method is used. A three-dimensional coordinate system showing the loads applied to the spinal construct and the 6 main motions (flexion, extension, left and right lateral bending, left and right axial rotation) were analyzed. The main motion is a rotation about the same axis as the applied moment. For each main motion, three main parameters were determined: neutral zone (NZ), elastic zone (EZ), and range of motion (ROM) (Figure 2). The neutral zone was the vertebral displacement at zero load with respect to the neutral position. The elastic zone was the vertebral displacement due to the load increase, from zero to the maximum load value. The ROM was the sum of the neutral and elastic zone.
However, the URES is the resultant without using reference geometry, and maximum URES were noted by recording the maximum resultant displacement. Because of the inherent coupling of motion in the lumbar spine rotational displacements were measured and compared by using the resultant motion in each of the planes at the application of the maximum moment.
Bresnahan et al.[30] observed an approximately 2.5-fold increase in left axial rotation at impaired segment occurred after the open (radical laminectomy) and interlaminar procedure (midline laminectomy with bilateral laminotomy and without removal of the posterior elements such as the spinous process, supra- and interspinous ligaments). Motion also increased by 1.3 times that of the intact as a result of the MEDS (Microendoscopic Decompression for Stenosis) procedure. Right axial rotation was most affected after the open and interlaminar procedure with a 2.1- and 1.6-fold increases in segmental motion at impaired segment. In our study, the maximum differences in URES between the LE model (conventional midline laminectomy with bilateral laminotomy and removal of posterior elements) and the intact model were 109.97% for axial rotation, 69.01% for extension, 48.83% for flexion, and 31.20% for lateral bending. It is important to note that ROM and displacement are distinct concepts, with differing ranges of change in ROM and maximum URES. However, despite these differences, the overall trend of change remains consistent.
Figure 1: A three-dimensional coordinate system showing the loads applied to the spinal construct and the 6 main motions (flexion, extension, left and right lateral bending, left and right axial rotation) were produced. The Figure coms from the Spine. 1990; 15(11):1142-1147
Figure 2. A typical physiologic load application and motion measurement procedure. Three motion parameters were determined: NZ = neutral zone; EZ = elastic zone; ROM = range-of-motion. The Figure coms from the Spine. 1990; 15(11):1142-1147.
10) Why stress distribution fields weren't shown? | Please revise;
stress distribution fields
Answer:
Thank you for your careful review and constructive suggestions regarding our manuscript.
Mobility of the impaired or restored segments should be taken into account in case of implantation with internal fixation systems, as well as on the development of new designs. Having these findings in mind, it is clear that the biomechanical stability of miniplates fixation in restorative laminoplasty need to be further tested, both clinically and numerically, before being considered for common use. It is essential to note that increased range of motion and displacement may result in nerve root impingement and narrowing of the foramen. Daily activities often exert forces on the lumbar spine, and excessive mobility in this area can lead to spinal canal narrowing and reduced neural foraminal area.
Furthermore, it is widely recognized that the spine is influenced by various load conditions, among which the invariable rule is joint movement. The movement of one segment corresponds to the mutual movement of another segment, which is a necessary prerequisite for understanding the interaction among intact spine, laminectomy, and laminoplasty. The goal of restorative laminoplasty was to integrate the posterior column complex, and thus the range of motion (ROM) and displacement of segmental vertebrae could directly reflect the biomechanical stability, efficacy, and feasibility of the surgery, as well as the stability of fixed segments and the risk of complication. This study was a preliminary assessment of the biomechanical stability, efficacy, and feasibility of restorative laminoplasty with miniplates fixation following laminectomy. The focus of our study was on comparing the biomechanical effects of load motion and displacement across different restorative laminoplasty procedures.
In order to fully understand the value of miniplate fixation in restorative laminoplasty, it is essential to thoroughly examine the characteristics of miniplate fixation from multiple perspectives. Additionally, analyzing the stress distribution fields within the internal fixation system is crucial for assessing the safety, effectiveness, and feasibility of restorative laminoplasty with miniplate fixation post-laminectomy. Current research on spine stress distribution analysis is predominantly focused on two-dimensional stress analysis, which provides a limited view of stress value distribution in different parts of the spine. However, the anatomical structure and movement of the spine is clinically intricate, making simple two-dimensional stress analysis insufficient in accurately depicting the true mechanical distribution of the spine at rest and during movement. Given that the spine naturally undergoes inevitable motion effects, studying biomechanics during spine movement holds great significance, albeit posing considerable challenges in spine biomechanics research.To address this, we propose that studying the biomechanics of the spine at rest and in motion can be achieved through three-dimensional and four-dimensional spine imaging, allowing for a better insight into the true internal mechanical distribution within the spine. Consequently, our future research will focus on spinal spatial motion mechanics. We have the necessary software and laboratory support in place, with preliminary experiments already planned.
The variation in our results may be primarily attributed to the effects of spatial motion mechanics. THMs exhibited a nearly-monohedral fixation, LSMs showed a nearly-pyramidal fixation, and HSMs displayed a nearly-prismatic fixation. Prismatic fixation is considered more effective than pyramidal fixation and monohedral fixation in avoiding rotation and displacement. This is attributed to its superior stress distribution in coplanar and coaxial configurations, thereby reducing instability caused by rotation and displacement. Moreover, the use of two-hole plates, like THMs in our study, with two points of stress have a limited anti-rotation effect and are prone to fracture retraction, collapse, and door-closing effect. This limitation becomes more pronounced in wider fractures, such as those at the lamina site. On the other hand, the triangular configuration fixed plates, such as LSMs in our study, are designed based on the principle of triangular stress points, offering excellent stabilization for axial movement, sagittal movement, forward and backward movement and lateral bending movement. However, during rotational activities, its stability is compromised due to the axial single surface and strip fixation, making it prone to deformation. In contrast, the four-point stressed plates like HSMs in our study, are fixed following a four-point spatial stress principle, enabling the formation of multiple triangles and multi-facet fixation in the axial direction when rotating, providing better stability during rotation.
The analysis presented above is a hypothesis drawn from the findings of our study, and further research is required to validate these conclusions, which is also a focus of our research group.
11) Figure 4 presentation needs to be rethink| Please revise;
Answer:
The Figure 4 is the URES distributions in the five models under all directions. Red represents large values and blue represents small values. The color bar on the left displays the corresponding value of the detail. The maximum URES corresponds to the highest value on the color bar.
12) The results discussion is an assembly between a revision work and the results presentation obtained here by FEA | Please revise;
Answer:
The issues have been revised as requested.
Suggestion: rethink the paper title, a more “clean” will be welcome!!
Answer:
The paper title has been revised to “Comparative Biomechanical Stability of Different Miniplates Fixation in Restorative Laminoplasty after laminectomy: A Finite Element Study”

Round 2
Reviewer 2 Report
Comments and Suggestions for Authors
I am satisfied that my comments have been addressed by the authors in a revised version.
Comments on the Quality of English LanguageQuality of English language acceptable.
Reviewer 4 Report
Comments and Suggestions for Authors
Nice work by the authors in addressing my comments. The paper looks good now